# A cross-national analysis of childhood predictors of daily smoking in adulthood

Sung Joon Jang [1,2] ✉, Pedro A. de la Rosa [3], R. Noah Padgett[4,5], Matt Bradshaw[1],
Tyler J. VanderWeele [4,5] & Byron R. Johnson[1,2,4]

## Abstract

**Background** Prior research on childhood predictors of cigarette smoking tends to focus on the prevalence rather than the quantity of smoking and rarely examined these predictors separately for smokers. Also scarce is cross-national research synthesizing the effects of childhood predictors.
**Methods** Using survey data from the Global Flourishing Study of 202,898 adults, weighted to be nationally representative of populations in 21 countries and one territory, we created continuous and binary measures of daily cigarette consumption in adulthood. The binary measure of daily smoking was regressed on childhood and demographic variables for the total sample, and the continuous measure was analyzed for the sample of smokers.
**Results** Random effects meta-analysis provides evidence that childhood maternal and paternal relationship quality (both total and smoker samples), religious service attendance (smoker sample), and being foreign-born (both samples) predict a lower likelihood of adult smoking, whereas being raised by a divorced parent (total sample), having been abused and/or an outsider in the family (both samples), and poor health growing up (both samples) predict a higher likelihood. Although effects are generally weak and mixed in some cases, their direction and strength tend to be consistent between the two samples as well as alternative measures of smoking with some exceptions. Overall, our findings are moderately robust against potential unmeasured confounding, while the effect sizes vary across countries.
**Conclusions** The present study offers an important new set of global findings based on a large-scale cross-national study of daily cigarette smoking and country-specific variations.

## Plain language summary

Childhood experiences have been studied in various countries to predict the proportion of adult cigarette smokers, but researchers have rarely examined childhood predictors in relation to the quantity of smoking among cigarette users. We analyzed survey data from the Global Flourishing Study of 202,898 adults, a nationally representative sample of approximately half of the world's population. Analyses reveal that prosocial childhood experiences (e.g., good parental relationships and religious service attendance) tend to predict a lower chance of daily cigarette smoking in adulthood. On the other hand, adverse experiences (e.g., parental divorce, abuse, being an outsider in the family, and poor health) predict a higher chance. The present study provides initial global evidence that childhood experiences affect health-risk behaviors in adulthood.

The age-standardized prevalence of current or former use of any smoked tobacco product among persons aged 15 or over decreased by 29.5% globally between 1990 and 2019, but only 4 out of 10 (39.7%) countries saw a significant decrease among young people (below age 26) in the past 30 years[1]. While the same trend (a 33.6% decrease) was observed for current cigarette smoking among adults (18 years or older) in the U.S. between 2002 and 2019[2], 88% of American adults who use cigarettes daily, report that they began smoking by age 18[https://www.cdc.gov/tobacco/data_statistics/fact_sheets/youth_data/tobacco_use/index.htm][3]. This pattern is similar to cigarette smokers in other industrial countries who mostly start smoking as teenagers[4]. Since young people are more susceptible to nicotine-dependence than adults[https://www.cdc.gov/tobacco/data_statistics/fact_sheets/youth_data/tobacco_use/index.htm], those who smoke regularly in their teenage years may become addicted to nicotine and continue to smoke into adulthood[3]. Given that the onset of smoking early in life—especially in teenage years—is a strong predictor of cigarette use in adulthood, it is important to study early predictors of smoking initiation in adolescence. Thus, this study examines childhood predictors—social relational and structural factors—of daily use of cigarettes in adulthood, analyzing new cross-national data to synthesize their effects on daily smoking in multiple countries, a feature largely missing in previous research.

Prior research on childhood social relational factors documents that parent-child relationships are associated with the child's likelihood of smoking: specifically, child attachment to parents and parental involvement

[1]Institute for Studies of Religion, Baylor University, Waco, TX, USA. [2]School of Public Policy, Pepperdine University, Malibu, CA, USA. [3]Institute for Culture and Society, Universidad de Navarra, Pamplona, Spain. [4]Human Flourishing Program, Institute for Quantitative Social Science, Harvard University, Cambridge, MA, USA. [5]Department of Epidemiology, Harvard T.H. Chan School of Public Health, Boston, MA, USA. ✉e-mail: Sung_Joon_Jang@baylor.edu

in the child's school, as well as parental monitoring and supervision, tend to be inversely related to smoking initiation in adolescence[5–8]. Another predictor of adolescent smoking is childhood maltreatment. For example, it was found that all types of childhood abuse (verbal, physical, and sexual) were associated with having ever smoked in life and heavy smoking in adulthood[9–11]. Also, a latent class growth analysis showed that physical abuse during early childhood (ages 3–5) predicted sharply increasing cigarette use during adolescence (ages 12–18), whereas neglect during the same period was related to a trajectory of gradually increasing cigarette use in adolescence[12]. A recent study found small but robust effects of various types of childhood maltreatment on "ever smoked" in adulthood[13]. In addition, prior research documents that adolescent involvement in religious activities, particularly religious service attendance, was inversely related to smoking in adolescence[3,7,14].

Childhood social structural factors that have been examined to predict adolescent smoking include family structure (parental marital status) and socioeconomic status (SES). Based on a cohort study in the U.S., parental divorce or separation was found to be positively related to smoking initiation by age 14 and current smoking in adulthood[11], whereas a longitudinal study of children in New Zealand showed that experiencing parental separation and having a single parent as well as low SES at age 7 predicted daily smoking by age 15[9,15,16]. A British birth cohort study[4] found that childhood SES, measured in terms of parental occupation, predicted persistent smoking in adulthood[5,6]. A prospective study in Australia also found a child's SES, measured by family income at birth, was related inversely to smoking at age 14, controlling for a child's age and sex, parental education, and maternal smoking through pregnancy[17]. However, the relationship became non-significant when adjusted for family income at age 14, implying an indirect effect of family income in early childhood on smoking in adolescence. In addition, controlling for parental smoking at age 14 as well as a child's behavioral problems and cognitive functions at ages 5 and 14, a child's smoking at age 14 was predicted by alternative measures of a child's SES, paternal and, to a greater extent, maternal education before a child's birth.

Although prior research has examined various childhood predictors of smoking initiation in childhood and adolescence or current smoking in adulthood using diverse methods and samples in different countries, they have relied mostly on a binary measure of smoking (e.g., 0 = non-smoker, 1 = smoker) and rarely studied quantity, such as number of cigarettes smoked daily, which is a key predictor of disease risk[1]. Further, despite the availability of various cross-national data sources [https://www.healthdata.org/research-analysis/gbd], synthesizing the effects of childhood predictors on smoking in different countries remains largely neglected. To address these oversights in prior research, utilizing new data from the Global Flourishing Study (GFS), we meta-analyze country-specific effects of (1) childhood predictors on the prevalence (i.e., binary measure) of daily cigarette smoking in adulthood in a sample of smokers and non-smokers combined and (2) those predictors on the quantity of daily cigarette consumption among smokers.

Our study is nested within a broader group of studies investigating childhood predictors of human flourishing, conceptualized as doing or being well in six domains of human life: happiness and life satisfaction, mental and physical health, meaning and purpose, character and virtue, close social relationships, and financial and material stability[18]. This paper is one of a collection of studies conducting analyses in parallel, aiming to be as consistent as possible across all studies with analytic methods so that any observed differences can be attributed to differences in constructs, not differences in analytic choices. The linked methods are specifically crafted to allow for a panoramic view of childhood predictors of flourishing in adulthood. The choice of these indicators for use in the GFS survey was a multi-phase process[19], and the selected items were chosen based largely on theory and in consultation with collaborators at Gallup. Specifically, the childhood predictor questions in the survey were selected based on prior research on longitudinal associations of childhood factors with subsequent well-being, including health. These factors include what prior research indicates have beneficial relationships with subsequent well-being (e.g.,

good relationship with parents, religious service attendance, and financial security) as well as two constructs of adverse childhood experiences, threat (abuse) and neglect (feeling like an outsider in the family). However, multicollinearity issues led to some conceptually distinct variables being omitted or modified (though issues may still occur for specific outcomes and countries), and thus future work may benefit from exploring specific factors more deeply.

In this exploratory study, we address the following research questions, all of which were pre-registered with the Center for Open Science (COS) (https://osf.io/6umhp).

Research Question #1: How do different aspects of a child's upbringing (Age [Year of Birth], Gender, Marital Status/Family Structure, Age 12 Religious Service Attendance, Relationship with Mother, Relationship with Father, Outsider Growing Up, Abuse, Self-Rated Health Growing Up, Immigration Status, Subjective Financial Status of Family Growing Up, Race/Ethnicity [when available], and Religious Affiliation at Age 12) predict daily smoking in adulthood?

Research Question #2: Do these associations vary by country?

Research Question #3: Are the observed relationships robust to potential unmeasured confounding, as assessed by E-values?

While exploring the first question, we anticipate the following relationships between demographic as well as childhood predictors and adult daily smoking based on prior research.

According to the most recent World Health Organization (WHO) report[20], the 2022 prevalence of smoking tobacco increased between ages 15–24 (13.3%) and 45–54 (26.4%) and declined thereafter through ages 85 or older (12.9%), showing a curvilinear relationship due partly to age differences in health-risk perception and psychological addiction[21,22]. The 2022 rate was higher among males (25.5%) than females (4.4%), likely as a result of traditional sex roles and gender socialization[23,24], while the difference tends to be greater in developing than developed countries. Thus, we expect these age and gender differences in daily cigarette smoking.

Good relationships with mother and father are expected to be inversely related to daily smoking in adulthood, whereas parents being not married, difficult financial status of family, abuse, and outsider in family growing up are expected to be positively associated with daily smoking[5–13]. Although religious involvement in childhood has rarely been studied in relation to smoking later in life, we anticipate an inverse relationship between religious service attendance in childhood and daily smoking in adulthood, as the former is likely to be positively associated with religious service attendance in adolescence, which has been found to be inversely related to the latter[3,7,14]. We also explore two other understudied factors with no particular expectation regarding the direction of associations: physical health and immigration status.

While physical health in childhood has rarely been studied in relation to smoking in adulthood, a longitudinal study in Sweden found an inverse relationship between self-rated health at ages 12–13 and smoking at ages 17–18 (the poorer health, the more likely to smoke) in a bivariate analysis, but the relationship turned non-significant in a multivariate analysis[25]. The initial relationship might have been explained by control variables, such as low self-esteem and less negative attitudes towards smoking, which were likely to be positively related to both poor health and smoking. Alternatively, however, children with poor physical health may avoid smoking out of health concerns compared to those with good health who may be more willing to experiment with smoking in adolescence and continue into adulthood.

Immigration status has also rarely been studied as a childhood predictor of smoking in adolescence or adulthood, but a systematic review of 15 studies on smoking among youth (ages 11–29) in European countries yielded mixed findings. That is, while some studies reported lower prevalence rates of smoking among migrant than native-born youth, others found the opposite or no difference[26] perhaps due in part to moderators, like the country of birth, sex, and duration of residence[27–30]. Positively stated, these mixed findings are consistent with the "immigrant paradox" whereby immigrant status can be both a risk and protective factor for health-risk

behavior. That is, immigration is a stressful process for adolescents and young adults facing various challenges in a new country, elevating the risk of smoking to cope with acculturative stress, whereas ethnic pride, adherence to traditional family values, and religiosity may reduce the likelihood of smoking.

Next, we explore whether the strength and even the direction of associations between childhood and demographic factors and adult smoking vary by country, reflecting the influence of diverse sociocultural, economic, and health contexts that characterize each nation. Finally, we examine the robustness of the observed associations between childhood factors and adult smoking against potential unmeasured confounding.

Results show that prosocial childhood experiences (e.g., good maternal and paternal relationships and religious service attendance) predict a lower likelihood of daily smoking in adulthood, whereas adverse experiences (e.g., parental divorce, abuse, and poor health) predict a higher likelihood. Although effects are generally weak and mixed in some cases, their direction and strength tend to be consistent between total and smoker samples and between binary and continuous measures of smoking with some exceptions. Overall findings are moderately robust against potential unmeasured confounding, while the effect sizes vary across countries. This study offers an important new set of global findings based on a large-scale cross-national study of daily cigarette smoking and country-specific variations.

## Methods
The following description of our methods has been adapted from Vander-Weele et al.[31]. Further methodological details are available elsewhere[19,31–36].

### Data
The Global Flourishing Study (GFS) is a longitudinal study of over 200,000 adults (age 18 or older) from 22 geographically and culturally diverse countries, with nationally representative sampling within each country, concerning the distribution of determinants of well-being[37]. Data for Wave 1 were collected by Gallup, Inc., principally during 2023, while some countries began data collection in 2022[34]. Four additional waves of panel data on study participants will be collected annually from 2024 to 2027. A total of 202,898 individuals participated in Wave 1 survey in Argentina, Australia, Brazil, Egypt, Germany, Hong Kong, India, Indonesia, Israel, Japan, Kenya, Mexico, Nigeria, the Philippines, Poland, South Africa, Spain, Sweden, Tanzania, Türkiye, the United Kingdom, and the United States. These countries were selected to (1) maximize coverage of the world's population, (2) ensure geographic, cultural, and religious diversity, and (3) prioritize feasibility and existing data collection infrastructure. The study was reviewed and approved by the institutional review boards at Baylor University (IRB reference #1841317) and Gallup (IRB reference #2021-11-02). Informed consent was obtained from all participants, and further details are available elsewhere[36].

The precise sampling design to ensure nationally representative samples varied by country, and further details are available elsewhere[34,36]. Survey items included various aspects of well-being such as happiness and life satisfaction, physical and mental health, meaning and purpose, character and virtue, close social relationships, and financial and material stability[18], along with other demographic, social, economic, political, religious, personality, childhood, community, health, and well-being variables. The data that support the findings of this article are openly available on the Open Science Framework. The specific dataset used was Wave 1 non-sensitive Global data https://osf.io/sm4cd/ available February 2024 —March 2026 via preregistration and publicly from then onwards[35]. During the translation process, Gallup adhered to TRAPD model (translation, review, adjudication, pretesting, and documentation) for cross-cultural survey research (https://ccsg.isr.umich.edu/chapters/translation/overview/).

### Measures
**Outcome variable.** Daily smoking was measured by an item, asking "About how many cigarettes do you smoke each day, if any?" (0 = None/Do not smoke, 1 = one, 2 = two, … 97 = 97+). Analyzing the number of cigarettes per day is complex given the highly skewed nature of the variable. It is often analyzed using zero-inflated models to predict the likelihood of non-zero and then predict the magnitude of the non-zero component. We approximate this using a two-part analysis strategy. First, we dichotomized the number of cigarettes per day into a binary variable (0 = None/Do not smoke, 1 = 1+) to regress on childhood predictors to evaluate what childhood factors predict the likelihood of any smoking as an adult. Second, we created a subsample of the non-zero component (smoker sample, N = 38,290) and regressed the non-zero component of the number of cigarettes on the childhood predictors to evaluate what childhood factors predict differences in the intensity of smoking.

**Childhood predictors.** Relationship with mother during childhood was assessed with the question: "Please think about your relationship with your mother when you were growing up. In general, would you say that relationship was very good, somewhat good, somewhat bad, or very bad?" Responses were dichotomized to be very/somewhat good versus very/somewhat bad. An analogous variable was used for relationship with father. "Does not apply" response was treated as a dichotomous control variable for respondents who did not have a mother or father due to death or absence. Parental marital status during childhood was assessed with responses of married, divorced, never married, and one or both had died. Financial status was measured with: "Which one of these phrases comes closest to your own feelings about your family's household income when you were growing up, such as when YOU were around 12 years old?" Responses were lived comfortably, got by, found it difficult, and found it very difficult. Abuse was assessed with yes/no responses to "Were you ever physically or sexually abused when you were growing up?" Participants were asked about being an outsider growing up: "When you were growing up, did you feel like an outsider in your family?" Childhood health was assessed by: "In general, how was your health when you were growing up? Was it excellent, very good, good, fair, or poor?" Immigration status was assessed with: "Were you born in this country, or not?" We will use this item as a proxy of childhood immigration status, which indicates whether participants spent time from infancy onwards in a country other than the one in which they are currently living. Religious service attendance during childhood was assessed with: "How often did YOU attend religious services or worship at a temple, mosque, shrine, church, or other religious building when YOU were around 12 years old?" with responses of at least once/week, one-to-three times/month, less than once/month, or never.

**Demographic variables.** Age (year of birth) was classified as 18–24, 25–34, 35–44, 45–54, 55–64, 65–74, 75–84, and 85 or older, the same categories that the WHO uses in its global report on smoking except the minimum age being 18, instead of 15[20]. Gender was assessed as male, female, and other. Childhood religious tradition/affiliation had response categories of Christianity, Islam, Hinduism, Buddhism, Judaism, Sikhism, Baha'i, Jainism, Shinto, Taoism, Confucianism, Primal/Animist/Folk religion, Spiritism, African-derived, some other religion, or no religion/atheist/agnostic with precise response categories varying by country[38]. Racial/ethnic identity was assessed in 19 of the 22 countries (except for Japan, Spain, and Sweden), and response categories were unique to each country. For additional details on the assessments, see the GFS codebook (https://osf.io/7uj6y/) or Crabtree et al.[31].

### Statistics and reproducibility
Descriptive statistics for the observed sample, weighted to be nationally representative within country, were estimated for each childhood predictor category. When smoker status was analyzed as a binary variable, a weighted modified Poisson regression model with complex survey adjusted standard errors was fit within each country for daily smoking on all of the aforementioned childhood predictors simultaneously, whereas when the number of cigarettes was analyzed, a weighted linear regression model with complex

## Table 1 | Frequency distributions of childhood predictors: Total and smoker samples

| Variable | Total sample (N = 202,898) | Smoker sample (N = 38,290) |
|---|---|---|
| **Relationship with mother** | | |
| Very good | 127,836 (63%) | 22,455 (59%) |
| Somewhat good | 52,439 (26%) | 10,699 (28%) |
| Somewhat bad | 11,060 (5.5%) | 2453 (6.4%) |
| Very bad | 4642 (2.3%) | 1033 (2.7%) |
| Does not apply | 5965 (2.9%) | 1433 (3.7%) |
| (Missing) | 956 (0.5%) | 217 (0.6%) |
| **Relationship with father** | | |
| Very good | 107,742 (53%) | 18,859 (49%) |
| Somewhat good | 55,714 (27%) | 10,979 (29%) |
| Somewhat bad | 15,807 (7.8%) | 3301 (8.6%) |
| Very bad | 8278 (4.1%) | 1816 (4.7%) |
| Does not apply | 13,985 (6.9%) | 3042 (7.9%) |
| (Missing) | 1372 (0.7%) | 294 (0.8%) |
| **Parental marital status** | | |
| Parents married | 152,001 (75%) | 27,665 (72%) |
| Divorced | 17,726 (8.7%) | 3755 (9.8%) |
| Parents were never married | 15,534 (7.7%) | 3232 (8.4%) |
| One or both parents had died | 7794 (3.8%) | 1586 (4.1%) |
| (Missing) | 9843 (4.9%) | 2052 (5.4%) |
| **Subjective financial status of family growing up** | | |
| Lived comfortably | 70,861 (35%) | 12,694 (33%) |
| Got by | 82,905 (41%) | 16,507 (43%) |
| Found it difficult | 35,852 (18%) | 6518 (17%) |
| Found it very difficult | 12,606 (6.2%) | 2413 (6.3%) |
| (Missing) | 674 (0.3%) | 160 (0.4%) |
| **Abuse** | | |
| Yes | 29,139 (14%) | 6334 (17%) |
| No | 167,279 (82%) | 30,385 (79%) |
| (Missing) | 6479 (3.2%) | 1571 (4.1%) |
| **Outsider growing up** | | |
| Yes | 28,732 (14%) | 6285 (16%) |
| No | 170,577 (84%) | 31,062 (81%) |
| (Missing) | 3589 (1.8%) | 943 (2.5%) |
| **Self-rated health growing up** | | |
| Excellent | 67,121 (33%) | 12,041 (31%) |
| Very good | 63,086 (31%) | 11,901 (31%) |
| Good | 47,378 (23%) | 9485 (25%) |
| Fair | 19,877 (9.8%) | 3795 (9.9%) |
| Poor | 4906 (2.4%) | 951 (2.5%) |
| (Missing) | 530 (0.3%) | 118 (0.3%) |
| **Immigration status** | | |
| Born in this country | 190,998 (94%) | 36,281 (95%) |
| Born in another country | 9791 (4.8%) | 1493 (3.9%) |
| (Missing) | 2110 (1.0%) | 517 (1.3%) |
| **Age 12 religious service attendance** | | |
| At least 1/week | 83,237 (41%) | 14,021 (37%) |
| 1–3/month | 33,308 (16%) | 6766 (18%) |
| <1/month | 36,928 (18%) | 7567 (20%) |
| Never | 47,445 (23%) | 9511 (25%) |
| (Missing) | 1980 (1.0%) | 425 (1.1%) |

Note. Smoker sample represents the subpopulation within each country that reported smoking at least one cigarette per day.

survey adjusted standard errors was fit. In the primary analyses, we used random effects meta-analyses of the regression coefficients[39,40] along with 95% confidence intervals (CIs), standard errors (continuous variable only), estimated proportions of effects across countries by thresholds (effect sizes larger than 0.1 and smaller than -0.1 for continuous variable with the thresholds being 1.1 and 0.9, respectively, for binary variable), heterogeneity of effect sizes "$\tau$" (the standard deviation of the distribution of effect sizes across the implied distribution with the 22 countries), and $I^2$ for evidence concerning variation within a given predictor category across countries[41]. Forest plots of estimates are available in the online supplement. Religious tradition/affiliation and race/ethnicity were used within country as control variables, when available, but these coefficients themselves were not included in the meta-analyses since categories/responses varied by country. All meta-analyses were conducted in $\mathbf{R}$[42] using the *metafor* package[43]. Within each country, a global test of association of each childhood predictor group with outcome was conducted, and a pooled $p$-value[44] across countries reported concerning evidence for association within any country. Bonferroni-corrected $p$-value threshold is provided based on the number of predictors[45,46]. All statistical tests are two-sided. For each predictor, we calculated E-value to evaluate the sensitivity of results to unmeasured confounding. An E-value is the minimum strength of the association an unmeasured confounder must have with both the outcome and the predictor, above and beyond all measured covariates, for an unmeasured confounder to explain away an association[47]. As a supplement, population-weighted meta-analyses of the regression coefficients were also estimated. All analyses were pre-registered with the COS prior to data access with two exceptions: only slight subsequent modification in the regression analyses due to multicollinearity and the regression analysis for the smoker sample (https://osf.io/3gupe/); all code to reproduce analyses are available in an online repository[32].

### Missing data
Missing data on all variables were imputed using multivariate imputation by chained equations, and five imputed datasets were used[48–51]. To account for variation in the assessment of certain variables across countries (e.g., religious tradition/affiliation and race/ethnicity), the imputation process was conducted separately in each country. This within-country imputation approach ensured that the imputation models accurately reflected country-specific contexts and assessment methods. The sampling weights were included as a variable in the imputation models to allow for specific variable missingness to be related to the probability of study inclusion.

### Accounting for complex sampling design
The GFS used different sampling schemes across countries based on the availability of existing panels and recruitment needs[34]. All analyses account for the complex survey design components by including weights, primary sampling units, and strata. For analyses involving the subpopulation of smokers, corrected weights were used to properly conduct domain estimation. Additional methodological detail, including accounting for the complex sampling design, is provided elsewhere[36,52].

### Reporting summary
Further information on research design is available in the Nature Portfolio Reporting Summary linked to this article.

## Results
### Descriptive statistics
Tables 1 and 2 report frequency distributions of the childhood predictors and demographic variables, first for the total sample of all 22 countries combined (N = 202,898) with the country sample sizes varying from 1473 in Türkiye to 38,312 in the United States, and then a reduced sample including only study participants who smoked at least one cigarette daily (N = 38,290). Instead of presenting the distribution of a continuous measure of daily

**Table 2 | Frequency distributions of demographic variables and participating country: Total and smoker samples**

| Variable | Total sample (N = 202,898) | Smoker sample (N = 38,290) |
|---|---|---|
| Age; year of birth | | |
| age 18–24; 1998–2005 | 27,007 (13%) | 3846 (10%) |
| age 25–34; 1988–1998 | 42,106 (21%) | 7893 (21%) |
| age 35–44; 1978–1988 | 36,980 (18%) | 8039 (21%) |
| age 45–54; 1968–1978 | 32,524 (16%) | 7095 (19%) |
| age 55–64; 1958–1968 | 29,400 (14%) | 6351 (17%) |
| age 65–74; 1948–1958 | 24,778 (12%) | 4101 (11%) |
| age 75–84; 1938–1948 | 8722 (4.3%) | 864 (2.3%) |
| 85 or older; 1938 or earlier | 1361 (0.7%) | 100 (0.3%) |
| (Missing) | 20 (<0.1%) | 2 (<0.1%) |
| Gender | | |
| Male | 98,411 (49%) | 24,063 (63%) |
| Female | 103,488 (51%) | 14,060 (37%) |
| Other | 602 (0.3%) | 81 (0.2%) |
| (Missing) | 397 (0.2%) | 87 (0.2%) |
| Country | | |
| Argentina | 6724 (3.3%) | 2293 (6.0%) |
| Australia | 3844 (1.9%) | 483 (1.3%) |
| Brazil | 13,204 (6.5%) | 2785 (7.3%) |
| Egypt | 4729 (2.3%) | 1085 (2.8%) |
| Germany | 9506 (4.7%) | 2590 (6.8%) |
| Hong Kong | 3012 (1.5%) | 956 (2.5%) |
| India | 12,765 (6.3%) | 1484 (3.9%) |
| Indonesia | 6992 (3.4%) | 2944 (7.7%) |
| Israel | 3669 (1.8%) | 882 (2.3%) |
| Japan | 20,543 (10%) | 4766 (12%) |
| Kenya | 11,389 (5.6%) | 713 (1.9%) |
| Mexico | 5776 (2.8%) | 1438 (3.8%) |
| Nigeria | 6827 (3.4%) | 368 (1.0%) |
| Philippines | 5292 (2.6%) | 1214 (3.2%) |
| Poland | 10,389 (5.1%) | 3230 (8.4%) |
| South Africa | 2651 (1.3%) | 669 (1.7%) |
| Spain | 6290 (3.1%) | 2082 (5.4%) |
| Sweden | 15,068 (7.4%) | 1858 (4.9%) |
| Tanzania | 9075 (4.5%) | 557 (1.5%) |
| Türkiye | 1473 (0.7%) | 789 (2.1%) |
| United Kingdom | 5368 (2.6%) | 928 (2.4%) |
| United States | 38,312 (19%) | 4176 (11%) |

Note. Smoker sample represents the subpopulation within each country that reported smoking at least one cigarette per day.

smoking, one that has a large range (97), we report its summary statistics in Supplementary Table S229. Specifically, the mean daily cigarette consumption was 2.0 (SD = 5.7) for the total sample and 11.3 (SD = 9.1) in the smoker sample. The mean of the total sample varied from 0.2 (SD = 1.8) for Tanzania to 9.8 (SD = 13.3) for Türkiye, whereas that of the smoker sample varied from 5.3 (SD = 6.6) for Kenya to 18.4 (SD = 13.1) for Türkiye. The binary measure showed that the prevalence of daily smoking was 18%, while the rate varied from 4.1% for Tanzania to 53% for Türkiye.

In the total sample, most of survey respondents had "very good" relationships with both mother (63%) and father (53%) and were raised by parents who were married (75%) in a family whose household income they felt had them "got by" (41%) when they were around 12 years old. Also, a large majority of respondents did not experience physical or sexual abuse (82%) or feel like an outsider in their family when they were growing up (84%). The highest percentage of the sample reported that they had "excellent" health growing up (33%) and attended religious services "at least 1/week" at the age of around 12 (41%), and participants were mostly native-born (94%), female (51%), and 25-34 years old (21%). The same categories were found to be modal in the smoker sample except for age (35–44, 21%) and gender (male, 63%). Missing data were less than 5%, with one exception being 5.4% missing on parental marital status in the smoker sample. While missingness varied by country as reported in supplementary tables for total sample (Supplementary Tables S1, S2, S5, S6, S9, S10, S13, S14, S17, S18, S21, S22, S25, S26, S29, S30, S33, S34, S37, S38, S41, S42, S45, S46, S49, S50, S53, S54, S57, S58, S61, S62, S65, S66, S69, S70, S73, S74, S77, S78, S81, S82, S85, and S86) and smoker sample (Supplementary Tables S139, S140, S143, S144, S147, S148, S151, S152, S155, S156, S159, S160, S163, S164, S167, S168, S171, S172, S175, S176, S179, S180, S183, S184, S187, S188, S191, S192, S195, S196, S199, S200, S203, S204, S207, S208, S211, S212, S215, S216, S219, S220, S223, and S224), it was generally rare with a few variables (e.g., parental marital status) having relatively high rates of missing in some countries. On average, missingness was not a concern.

**Total sample analysis**

Table 3 presents random effects meta-analytic estimates of associations between the childhood and demographic variables and the binary measure of daily smoking, converted to the risk ratio (RR) scale. Consistent with prior research[5–8], "good" childhood relationships with mother (0.94, 95% CI [0.90, 0.99]) and father (0.94, [0.89, 0.99]) were both inversely related to smoking at least one cigarette per day in adulthood (compared to the reference category of "bad" relationship), while multicollinearity was likely for maternal relationship quality. These meta-analyzed effects indicate that across countries, individuals who had good relationships with mother and father in childhood were, on average, 0.94 times less likely (i.e., a decrease of approximately 6%) to consume at least one cigarette daily when they became adults than those who reported bad parental relationships. It was interesting to see the average effect size of good relationships with mother and father was no different (0.94). According to "estimated proportion of effects by threshold" (which is based on the "calibrated effect estimates," not raw estimated effects)[33], the effect size of parental relationship quality did not vary widely across 22 countries, as all country-specific estimates of maternal relationship quality were found within the range of thresholds (i.e., $0.90 \leq RR \leq 1.10$). An exception was paternal relationship quality estimate that was found to be less than an RR of 0.90 in about a quarter (0.27) of those countries including Egypt, India, Israel, Nigeria, and Tanzania, which indicated stronger preventive effect of early paternal relationship quality on later smoking in those countries (see Supplementary Tables S15, S27, S35, S51, and S75). The pooled global p-value provided evidence against the null hypothesis that the effect size of maternal relationship quality is 0 in all 22 countries ($p = 0.045$), while it was not the case with paternal relationship quality ($p = 0.165$).

As found previously[9,11,15], parental marital status was also a significant childhood predictor of daily smoking in adulthood with global p-value ($p = 6.99\text{e-}06$) being below the Bonferroni correction threshold ($p < 0.004$). That is, individuals who had divorced (1.26, [1.20, 1.33]) or single/never married parent (1.16, [1.06, 1.27]) or one or both parents died (1.10, [1.03, 1.17]) growing up were more likely to smoke one or more cigarettes daily later in life, compared to those who had married parents. Specifically, having a divorced parent, a single/never married parent, and any parent died in childhood were, on average, 1.26, 1.16, and 1.10 times more likely (an increase of about 26%, 16%, and 10%, respectively) to result in daily smoking in adulthood than being raised by married parents. It is worth noting that the effect of being raised by a divorced parent on daily smoking in adulthood was universal, as its calibrated effect estimate was above an RR of 1.10 in all GFS countries (1.00). Also, the above-threshold effect of having a single/never married parent or any parent died in childhood was estimated in

**Table 3 | Random effects meta-analysis of regression of daily cigarette consumption (binary) on childhood predictors: Total sample (N = 202,898)**

| Variable | Category | RR | 95% CI | Estimated proportion of effects by threshold | | | Global p-value |
| | | | | <0.90 | >1.10 | $I^2$ | |
|---|---|---|---|---|---|---|---|
| Relationship with mother | (Ref: very bad/somewhat bad) | | | | | | 0.045* |
| | Very good/somewhat good | 0.94 | (0.90,0.99) | 0.00 | 0.00 | <0.1[a] | |
| Relationship with father | (Ref: very bad/somewhat bad) | | | | | | 0.165 |
| | Very good/somewhat good | 0.94 | (0.89,0.99) | 0.27 | 0.00 | 31.0 | |
| Parental marital status | (Ref: parents married) | | | | | | 6.99e-06** |
| | Divorced | 1.26 | (1.20,1.33) | 0.00 | 1.00 | 32.7 | |
| | Single, never married | 1.16 | (1.06,1.27) | 0.00 | 0.64 | 62.1 | |
| | One or both parents had died | 1.10 | (1.03,1.17) | 0.00 | 0.50 | 8.4 | |
| Subjective financial status of family growing up | (Ref: got by) | | | | | | 0.002** |
| | Lived comfortably | 1.00 | (0.97,1.04) | 0.00 | 0.00 | 29.4 | |
| | Found it difficult | 0.99 | (0.94,1.03) | 0.05 | 0.05 | 29.6 | |
| | Found it very difficult | 1.06 | (0.99,1.13) | 0.05 | 0.27 | 16.5 | |
| Abuse | (Ref: no) | | | | | | 2.35e-05** |
| | Yes | 1.26 | (1.19,1.34) | 0.00 | 0.95 | 64.1 | |
| Outsider growing up | (Ref: no) | | | | | | 1.54e-05** |
| | Yes | 1.20 | (1.14,1.28) | 0.00 | 0.82 | 59.8 | |
| Self-rated health growing up | (Ref: good) | | | | | | 1.10e-05** |
| | Excellent | 1.03 | (0.98,1.10) | 0.09 | 0.27 | 60.8 | |
| | Very good | 0.99 | (0.95,1.04) | 0.09 | 0.05 | 42.2 | |
| | Fair | 0.97 | (0.92,1.02) | 0.05 | 0.00 | 12.5 | |
| | Poor | 0.52 | (0.17,1.55) | 0.45 | 0.23 | 99.4 | |
| Immigration status | (Ref: born in this country) | | | | | | 1.77e-05** |
| | Born in another country | 0.90 | (0.75,1.10) | 0.50 | 0.23 | 85.7 | |
| Age 12 religious service attendance | (Ref: never) | | | | | | 2.19e-05** |
| | At least 1/week | 0.96 | (0.90,1.03) | 0.32 | 0.14 | 56.3 | |
| | 1–3/month | 1.05 | (0.95,1.16) | 0.14 | 0.36 | 75.7 | |
| | <1/month | 1.00 | (0.95,1.05) | 0.05 | 0.05 | 30.5 | |
| Age; year of birth | (Ref: age 18–24; 1998–2005) | | | | | | 1.44e-06** |
| | age 25–34; 1988–1998 | 1.39 | (1.27,1.53) | 0.00 | 0.95 | 73.6 | |
| | age 35–44; 1978–1988 | 1.54 | (1.36,1.75) | 0.00 | 0.95 | 85.8 | |
| | age 45–54; 1968–1978 | 1.51 | (1.31,1.74) | 0.00 | 0.95 | 87.3 | |
| | age 55–64; 1958–1968 | 1.53 | (1.30,1.80) | 0.00 | 0.86 | 87.8 | |
| | age 65–74; 1948–1958 | 1.14 | (0.92,1.42) | 0.32 | 0.45 | 87.8 | |
| | age 75–84; 1938–1948 | 0.22 | (0.04,1.29) | 0.64 | 0.18 | 99.5 | |
| | 85 or older; 1938 or earlier[b] | 0.04 | (0.00,0.37) | 0.68 | 0.27 | 99.5 | |
| Gender | (Ref: male) | | | | | | 1.53e-06** |
| | Female | 0.32 | (0.19,0.52) | 0.86 | 0.00 | 99.7 | |
| | Other[b] | 0.11 | (0.01,1.05) | 0.61 | 0.28 | 99.4 | |

* p < 0.05; ** p < 0.004 (Bonferroni corrected threshold).
[a]Estimate of heterogeneity is likely unstable. See our online Supplementary Figs. for more detail on the heterogeneity of effects.
[b]Group is very small (<0.1% of the observed sample) within several countries leading large uncertainty in this estimate or even complete separation—be cautious about interpreting this estimate;
CI = confidence interval; the Estimated proportion of effects is the estimated proportion of effects above (or below) a threshold based on the calibrated effect sizes[41]; $I^2$ is an estimate of the variability in means due to heterogeneity across countries vs. sampling variability; the Global p-value corresponds to the joint test of the null hypothesis that the country-specific joint parameter Wald tests (all parameters within variable groups are zero) are all null in all 22 countries; and additional details of heterogeneity of effects are available in our online Supplementary Figs.

about two thirds (0.64) and a half (0.50) of those countries, respectively, including the U.S., Nigeria, Sweden, Australia, the U.K., Poland, Tanzania, the Philippines, Türkiye, Egypt, Kenya, Japan, Brazil, and Indonesia (see Supplementary Figs. S4 and S5). Given these countries being geographically and culturally diverse, overall findings tend to indicate that the effect of parental marital status in childhood on daily smoking in adulthood is geography/culture independent.

While the global p-value of childhood family SES was significant (p = 0.002), two of its three categories—"lived comfortably" (1.00 [0.97, 1.04]) and "found it difficult" (0.99 [0.94, 1.03])—had little effect on daily smoking later in life, compared to the reference category of "got by." The exception was growing up in a family that had a "very difficult" financial status (1.06, [0.99, 1.13]), which increased the likelihood of daily smoking in adulthood by, on average, about 6% compared to the reference category. The

health-risk effect of extreme family financial difficulty was estimated to be above an RR of 1.10 in about a quarter (0.27) of countries including Nigeria, India, Poland, and the Philippines (Supplementary Tables S27, S51, S55, and S59 and Supplementary Fig. S8).

A relatively strong effect was found for childhood abuse on daily cigarette consumption in adulthood (1.26, [1.19, 1.34]) as individuals physically or sexually abused when they were growing up were found to be about 26% more likely to smoke daily than those who reported no such abuse in childhood. Similarly, having grown up feeling like an outsider in the family also increased the risk of daily use of cigarettes in adulthood, on average, by about 20% (1.20, [1.14, 1.28]). The health-risk effect of these childhood adverse experiences on later smoking was estimated to be above an RR of 1.10 in at least eight (0.82) or nine (0.95) out of 10 GFS countries with Poland and Israel being exceptions, where the effect was found to be weak to null (Supplementary Tables S35 and S59). This finding indicates that childhood abuse and neglect are universal risk factors for daily smoking in adulthood across geographically or culturally diverse countries.

Like childhood family SES, self-rated health growing up had a limited effect on daily smoking in adulthood. Compared to "good" health, "excellent" (1.03, [0.98, 1.10]), "very good" (0.99, [0.95, 1.04]), and "fair" (0.97, [0.92, 1.02]) health in childhood had a mixed and weak effect on daily smoking later in life, while "poor" health (0.52, [0.17, 1.55]) had a rather substantial effect, decreasing the likelihood of adult daily smoking by 48% compared to the reference category. Furthermore, the effect of poor health in childhood varied widely across countries, as it was estimated to be less than an RR of 0.90 in almost a half (0.45) of countries (Germany, Hong Kong, India, Indonesia, Israel, Kenya, South Africa, Spain, Tanzania, and the U.K.; Supplementary Tables S19, S23, S27, S31, S35, S43, S63, S67, S75, and S83) and above an RR of 1.10 in about a quarter (0.23) of them (Australia, Mexico, Sweden, Türkiye, and the U.S.; Supplementary Tables S7, S47, S71, S79, and S87). That is, while poor health in childhood was, on average, likely to decrease daily smoking in adulthood, it can increase the likelihood, depending on the country.

Regarding immigration status, foreign-born respondents were, on average, 0.90 times less likely to consume cigarette daily in adulthood (0.90, [0.75, 1.10]) compared to their native-born counterparts, as previously found based on adult immigrants[27–30]. The effect, however, varied as much as the effect of poor health in childhood did. Specifically, it was estimated to be less than an RR of 0.90 in a half (0.50) of countries (Argentina, Australia, Germany, Hong Kong, Nigeria, South Africa, Spain, Tanzania, Türkiye, the U.K., and the U.S.; Supplementary Tables S3, S7, S19, S23, S51, S63, S67, S75, S79, S83, and S87) and above an RR of 1.10 in about a quarter (0.23) of them (India, Japan, Mexico, the Philippines, Poland, and Sweden; Supplementary Tables S27, S39, S47, S55, S59, and S71), as Supplementary Fig. S15 shows. This finding suggests that being an immigrant can be both a risk and protective factor for daily smoking, consistent with the "immigrant paradox"[26].

We found mixed and weak effect of childhood religious service attendance on adult daily smoking. Specifically, the anticipated protective effect, though weak, was confined to attending religious services "at least once a week" in childhood (.096, [0.90, 1.03]), which decreased the likelihood of daily smoking in adulthood by about 4% compared to having "never" attended. The effect of at least weekly religious service attendance was estimated to be below an RR of 0.90 in about one third (0.32) of those 22 countries, whereas it was to be above an RR of 1.10 in about one (0.14) out of 10 GFS countries with the effect being small or little in the remaining (54%) countries. While the effect varied rather widely, this result is consistent with the previous finding that religious service attendance tended to have an inverse relationship with smoking when the frequency of attendance was weekly or more than once a week[3,7,14]. Conversely, however, attending religious services "1–3 times a month" (1.05, [0.95, 1.16]) was found to increase the likelihood of adult smoking, whereas attending services "less than once a month" (1.00, [0.95, 1.05]) had little effect on smoking.

Controlling for childhood predictors, the effect of age/year of birth differs across age categories in a possibly nonlinear pattern with the largest

effect being found for 35-44 (1.54, [1.36, 1.75]), followed by age 55-64 (1.53, [1.30, 1.80]), age 45-54 (1.51, [1.31, 1.74]), 25-34 (1.39, [1.27, 1.53]), 65-74 (1.14, [0.92, 1.42]), 18-24 (the reference category), 75-84 (0.22, [0.04, 1.29]), and 85 or older (0.04, [0.00, 0.37]). This is consistent with the most recent WHO report[20] except that the age groups of 35-44 and 55-64 switched their positions. The oldest category's estimate, however, should be interpreted with caution because the age group is very small (<0.1% of the observed sample) within several countries (Egypt, the Philippines, South Africa, and Spain). Females were, on average, 0.32 times less likely (a 68% decrease) to report daily cigarette consumption in adulthood (0.32, [0.19, 0.52]) than males[20], whereas individuals of "other" gender were also 0.11 times less likely (a 89% decrease) to consume cigarettes daily (0.11, [0.01, 1.05]) compared to the reference category of being male. The latter finding, however, should be interpreted with caution, as 12 of 22 countries had either no respondent (Egypt, India, Tanzania, and Türkiye) or less than 0.1% of the sample (Hong Kong, Indonesia, Israel, Kenya, Mexico, Nigeria, Poland, and South Africa) in that gender category.

In sum, overall results for the binary measure of daily smoking were generally consistent with previous findings based on the prevalence of smoking[4,7–11,26].

To assess the sensitivity or robustness of the meta-analytic effect estimates to potential unmeasured confounding, we report E-values of the above estimates (and those of estimates from smoker-sample analyses presented below) in Table 4. For example, an unmeasured confounder that was associated with both the exposure to childhood abuse and daily cigarette consumption by risk ratios of 1.84 each, above and beyond the covariates already adjusted for, could fully explain away the association, but weaker joint confounder associations could not. To shift the 95% CI to include the null, an unmeasured confounder that was associated with both the exposure and daily smoking by risk ratios of 1.66 each could suffice, but weaker joint confounder could not. While E-values suggest that estimated effects are, in general, moderately robust to unmeasured confounding, those close to 1.00 indicate very little unmeasured confounding would be required. For instance, the effect of "lived comfortably" growing up had E-value of 1.07 and may be explained away by an unmeasured confounder, such as parental education that is related positively to family financial status and inversely to parental smoking, which in turn is positively associated with a child's smoking in adolescence and adulthood[53–56]. Thus, these effect estimates should be interpreted with their limited robustness in mind.

## Smoker sample analysis

Table 5 shows results from meta-analyzing effects of the same predictors on the quantity of daily cigarette consumption among daily smokers. In terms of the direction of relationship, consistent results were found for the effects of maternal and paternal relationship quality, family SES (except for the "lived comfortably" category), abuse, outsider, and self-rated health (except for the "fair" health category) growing up as well as immigration status and gender on the number of cigarettes smoked daily in adulthood. That is, as found in the binary analysis for the total sample, parental relationship quality, "difficult" family finance, "very good" and "poor" health, being foreign-born, and being female were inversely related to the quantity of daily cigarette consumption, whereas "very difficult" family finance, abuse, outsider, and "excellent" health growing up were positively related to daily smoking. For example, daily smokers who had good relationships with mother (−0.34, [−1.04, 0.35]) and father in childhood (−0.38, [−0.81, 0.05]) reported smoking, on average, 0.34 and 0.38 cigarettes less per day than those who had bad relationships with them, and childhood abuse increased the quantity of daily smoking by almost a half (0.48) cigarette per day compared to having no such adverse childhood experience. In addition, as found in the binary analysis, some variables' categories had mixed relationships with the quantity of daily smoking. For instance, two categories of childhood health above—"excellent" (0.08, [−0.54, 0.70]) and "very good" (−0.09, [−0.60, 0.42])—and below the reference category of "good" health—"fair" (0.29, [−0.25, 0.82]) and "poor" health (−0.24, [−1.15, 0.68])—had both positive and negative relationships with daily

**Table 4 | Sensitivity of meta-analyzed childhood predictors to unmeasured confounding**

| Variable | Category | Total sample (N = 202,898) Binary | | Smoker sample (N = 38,290) Continuous | |
|---|---|---|---|---|---|
| | | E-value | E-value limit | E-value | E-value limit |
| Relationship with mother | (Ref: very bad/somewhat bad) | | | | |
| | Very good/somewhat good | 1.31 | 1.12 | 1.31 | 1.00 |
| Relationship with father | (Ref: very bad/somewhat bad) | | | | |
| | Very good/somewhat good | 1.32 | 1.11 | 1.34 | 1.00 |
| Parental marital status | (Ref: parents married) | | | | |
| | Divorced | 1.84 | 1.68 | 1.29 | 1.00 |
| | Single, never married | 1.59 | 1.31 | 1.05 | 1.00 |
| | One or both parents had died | 1.42 | 1.20 | 1.04 | 1.00 |
| Subjective financial status of family growing up | (Ref: got by) | | | | |
| | Lived comfortably | 1.07 | 1.00 | 1.22 | 1.00 |
| | Found it difficult | 1.13 | 1.00 | 1.13 | 1.00 |
| | Found it very difficult | 1.31 | 1.00 | 1.38 | 1.00 |
| Abuse | (Ref: no) | | | | |
| | Yes | 1.84 | 1.66 | 1.39 | 1.05 |
| Outsider growing up | (Ref: no) | | | | |
| | Yes | 1.70 | 1.53 | 1.19 | 1.00 |
| Self-rated health growing up | (Ref: good) | | | | |
| | Excellent | 1.22 | 1.00 | 1.13 | 1.00 |
| | Very good | 1.11 | 1.00 | 1.14 | 1.00 |
| | Fair | 1.20 | 1.00 | 1.28 | 1.00 |
| | Poor | 3.26 | 1.00 | 1.25 | 1.00 |
| Immigration status | (Ref: born in this country) | | | | |
| | Born in another country | 1.45 | 1.00 | 1.28 | 1.00 |
| Age 12 religious service attendance | (Ref: never) | | | | |
| | At least 1/week | 1.24 | 1.00 | 1.55 | 1.30 |
| | 1–3/month | 1.29 | 1.00 | 1.73 | 1.50 |
| | <1/month | 1.05 | 1.00 | 1.47 | 1.24 |
| Age; year of birth | (Ref: age 18–24; 1998–2005) | | | | |
| | age 25–34; 1988–1998 | 2.13 | 1.85 | 1.89 | 1.69 |
| | age 35–44; 1978–1988 | 2.46 | 2.05 | 2.62 | 2.22 |
| | age 45–54; 1968–1978 | 2.39 | 1.95 | 2.85 | 2.37 |
| | age 55–64; 1958–1968 | 2.44 | 1.93 | 3.19 | 2.56 |
| | age 65–74; 1948–1958 | 1.54 | 1.00 | 3.04 | 2.38 |
| | age 75–84; 1938–1948 | 8.46 | 1.00 | 2.14 | 1.38 |
| | 85 or older; 1938 or earlier[a] | 55.48 | 4.83 | 1.91 | 1.00 |
| Gender | (Ref: male) | | | | |
| | Female | 5.77 | 3.26 | 2.06 | 1.55 |
| | Other[a] | 18.20 | 1.00 | 3.10 | 2.09 |

[a]Group is very small (<0.1% of the observed sample) within several countries leading to high uncertainty in this estimate—be cautious about interpreting this estimate; E-value is the minimum strength of the association an unmeasured confounder must have with both the outcome and the predictor, above and beyond all measured covariates, for an unmeasured confounder to explain away an association[47].

smoking each. These results were generally consistent in terms of the strength of relationship as well. That is, as we found above, childhood abuse had a relatively strong effect on daily smoking in adulthood, whereas maternal and paternal relationship quality, immigration status, outsider growing up, and the categories of "difficult" family SES as well as "excellent" and "very good" health were rather weakly related to adult daily smoking.

However, inconsistent results were found for the remaining two childhood predictors: parental marital status and religious service attendance. First, while parental marital status had a weak relationship with daily cigarette consumption as found earlier, one of the three categories had a

notable opposite effect on daily smoking, compared to what was found in the binary analysis. Although we found in the total sample that individuals who had a divorced parent growing up were somewhat more likely to smoke at least one cigarette daily later in life than those who had married parents, daily smokers with the same background reported smoking, on average, 0.31 cigarette less per day than the reference group (i.e., smokers who had married parents growing up). The average effect of being raised by a divorced parent was found to vary by 0.98 cigarettes per day across 22 countries, implying significant heterogeneity ($\tau$) in the effect among those countries. In addition, half (0.50) of countries (India, Hong Kong, the

**Table 5 | Random effects meta-analysis of regression of daily cigarette consumption (continuous) on childhood predictors: Smoker sample (N = 38,290)**

| Variable | Category | Est | 95% CI | SE | Estimated proportion of effects by threshold <−0.10 | Estimated proportion of effects by threshold >0.10 | Heterogeneity (τ) | $I^2$ | Global p-value |
|---|---|---|---|---|---|---|---|---|---|
| Relationship with mother | (Ref: very bad/somewhat bad) | | | | | | | | 0.201 |
| | Very good/somewhat good | −0.34 | (−1.04,0.35) | 0.36 | 0.64 | 0.36 | 1.15 | 57.5 | |
| Relationship with father | (Ref: very bad/somewhat bad) | | | | | | | | 0.166 |
| | Very good/somewhat good | −0.38 | (−0.81,0.05) | 0.22 | 0.68 | 0.18 | 0.49 | 25.6 | |
| Parental marital status | (Ref: parents married) | | | | | | | | 0.008* |
| | Divorced | −0.31 | (−0.91,0.29) | 0.31 | 0.50 | 0.36 | 0.98 | 54.0 | |
| | Single, never married | 0.01 | (−0.69,0.71) | 0.36 | 0.41 | 0.55 | 1.20 | 59.9 | |
| | One or both parents had died | −0.01 | (−0.58,0.57) | 0.29 | 0.00 | 0.00 | <.01[a] | <0.1[a] | |
| Subjective financial status of family growing up | (Ref: got by) | | | | | | | | 4.05e-05** |
| | Lived comfortably | −0.19 | (−0.56,0.17) | 0.19 | 0.45 | 0.32 | 0.57 | 47.7 | |
| | Found it difficult | −0.08 | (−0.42,0.26) | 0.17 | 0.00 | 0.00 | <.01[a] | <0.1[a] | |
| | Found it very difficult | 0.46 | (−0.19,1.11) | 0.33 | 0.23 | 0.73 | 0.87 | 32.9 | |
| Abuse | (Ref: no) | | | | | | | | 0.064 |
| | Yes | 0.48 | (0.01,0.96) | 0.24 | 0.19 | 0.81 | 0.71 | 44.7 | |
| Outsider growing up | (Ref: no) | | | | | | | | 0.048* |
| | Yes | 0.16 | (−0.33,0.64) | 0.25 | 0.36 | 0.59 | 0.75 | 46.5 | |
| Self-rated health growing up | (Ref: good) | | | | | | | | 0.007* |
| | Excellent | 0.08 | (−0.54,0.70) | 0.32 | 0.27 | 0.59 | 1.20 | 72.4 | |
| | Very good | −0.09 | (−0.60,0.42) | 0.26 | 0.41 | 0.50 | 0.84 | 56.8 | |
| | Fair | 0.29 | (−0.25,0.82) | 0.27 | 0.41 | 0.55 | 0.69 | 33.9 | |
| | Poor | −0.24 | (−1.15,0.68) | 0.47 | 0.38 | 0.52 | 1.23 | 38.5 | |
| Immigration status | (Ref: born in this country) | | | | | | | | 2.17e-05** |
| | Born in another country | −0.30 | (−2.04,1.45) | 0.89 | 0.64 | 0.32 | 3.80 | 91.6 | |
| Age 12 religious service attendance | (Ref: never) | | | | | | | | 0.044* |
| | At least 1/week | −0.80 | (−1.27,−0.32) | 0.24 | 0.95 | 0.00 | 0.49 | 20.1 | |
| | 1–3/month | −1.15 | (−1.61,−0.70) | 0.23 | 1.00 | 0.00 | 0.41 | 15.1 | |
| | <1/month | −0.63 | (−1.04,−0.23) | 0.21 | 0.95 | 0.00 | 0.34 | 13.5 | |
| Age; year of birth | (Ref: age 18–24; 1998–2005) | | | | | | | | 2.78e-06** |
| | age 25–34; 1988–1998 | 1.48 | (1.08,1.87) | 0.20 | 0.00 | 1.00 | 0.09 | 1.0 | |
| | age 35–44; 1978–1988 | 2.84 | (2.11,3.57) | 0.37 | 0.00 | 1.00 | 1.35 | 66.1 | |
| | age 45–54; 1968–1978 | 3.22 | (2.39,4.06) | 0.43 | 0.00 | 1.00 | 1.62 | 72.7 | |
| | age 55–64; 1958–1968 | 3.74 | (2.73,4.75) | 0.52 | 0.00 | 1.00 | 2.02 | 76.7 | |
| | age 65–74; 1948–1958 | 3.51 | (2.40,4.62) | 0.57 | 0.00 | 1.00 | 2.11 | 69.2 | |
| | age 75–84; 1938–1948 | 1.95 | (0.46,3.45) | 0.76 | 0.30 | 0.70 | 2.73 | 71.3 | |
| | 85 or older; 1938 or earlier[b] | 1.52 | (−0.21,3.24) | 0.88 | 0.13 | 0.87 | 2.14 | 43.8 | |
| Gender | (Ref: male) | | | | | | | | 3.18e-06** |
| | Female | −1.81 | (−2.83,−0.79) | 0.52 | 0.68 | 0.23 | 2.16 | 93.1 | |
| | Other[b] | −3.62 | (−5.38,−1.86) | 0.90 | 0.93 | 0.07 | 2.31 | 55.8 | |

* p < 0.05; ** p < 0.004 (Bonferroni corrected threshold).
[a]Estimate of heterogeneity is likely unstable. See our online Supplementary Figs. for more detail on heterogeneity of effects.
[b]Group is very small (<0.1% of the observed sample) within several countries leading large uncertainty in this estimate or even complete separation—be cautious about interpreting this estimate; CI confidence interval; the estimated proportion of effects is the estimated proportion of effects above (or below) a threshold based on the calibrated effect sizes[41]; $I^2$ is an estimate of the variability in means due to heterogeneity across countries vs. sampling variability; the Global p-value corresponds to the joint test of the null hypothesis that the country-specific joint parameter Wald tests (all parameters within variable groups are zero) are all null in all 22 countries; and additional details of heterogeneity of effects are available in our online Supplementary Figs.

Philippines, South Africa, Nigeria, Spain, the U.S., Mexico, Poland, Indonesia, and Türkiye) had the unexpected, preventive effects stronger than −0.10, while about one-third (0.36) of countries (Argentina, Australia, Egypt, Israel, Japan, Kenya, Sweden, the U.K) had the anticipated, health-risk effects stronger than 0.10 (Supplementary Fig. S57).

Next, the two categories of less-than-once-a-week religious service attendance at age 12, which had positive-to-little effect on the binary measure of smoking in the total sample, were found to be inversely related to the quantity of daily smoking, as expected. Specifically, daily smokers who had attended religious services "1–3 times" or "less than once a month" at age 12 reported smoking, on average, 1.15 and 0.63 cigarette less per day, respectively, than those who never attended religious services. Estimated proportions of all three attendance categories' effects below the threshold of −0.10 were very high (0.95, 1.00, and 0.95), where no effects estimated were above the threshold of 0.10 (Supplementary Figs. S70–S72), although we found in the binary analysis for the total sample that proportions of effects below and above the threshold were 0.32, 0.14, and 0.05 and 0.14, 0.36, and 0.05, respectively. This may suggest that childhood religious service attendance has more uniform effects on daily smoking in adulthood among smokers than in a total population. E-values (Table 4) suggest that many of the estimated effects are moderately robust to unmeasured confounding.

In sum, the childhood predictors tended to have the anticipated effects, though generally weak and mixed in some cases, on the quantity of daily cigarette consumption in the smoker sample, as was the case with their effects on the prevalence of daily smoking in the total sample, with some exceptions. We also conducted the same random effects meta-analysis for the total sample, but it is not presented as primary analysis since their estimates are likely biased to the extent that the model was misspecified, as the outcome variable is highly skewed with a large number of zeros (81% no daily smoking). Instead, we present the results along with sensitivity analysis (see Supplementary Tables S135 and S136).

### Comparative summary

Regression results with the binary measure of daily smoking for the total sample and the continuous measure for the smoker sample showed that adverse childhood experiences (ACE) had strong effects on cigarette smoking in adulthood relative to other childhood predictors: being raised by a divorced parent (compared to both parents married) on the binary measure, "very difficult" family financial status (compared to "got by" status) on the continuous measure, and being abused and feeling like an outsider in the family growing up on both measures. Also notable, however, was relative importance of two childhood factors—having "good" (compared to "bad") relationships with mother and father and attending religious services with any frequency (compared to "never") growing up—in predicting the quantity of adult daily cigarette consumption in the smoker sample. In fact, the effect of religious service attendance of any frequency in childhood was stronger than any ACE factors perhaps because this factor was less variant among smokers than in the total sample. In addition, we found that demographic variables tended to have larger effects on adult smoking than childhood predictors including ACE factors, regardless of the sample or measure of smoking analyzed, confirming the significant role age and gender play in explaining cigarette smoking[20].

### Supplemental analysis

To supplement our random effects meta-analysis, we conducted a set of population-weighted meta-analysis, where each country's results were weighted by its population size in 2023, equally treating each person in the 22 countries instead of each country. As a result, India, the largest country that constituted about 40% of all GFS country populations combined, received the most weight[33]. These alternative meta-analysis results were generally consistent with their random effects counterparts except for one predictor, reflecting greater weights given to the results of that largest country. The exception was immigration status: that is, being foreign-born—which was inversely associated with binary and continuous measures of daily smoking in the primary meta-analysis (Tables 3 and 5)—was found

to be positively related to both measures of daily smoking. These relationships reflect the increased weight given to the positive effect of being foreign-born on daily smoking, observed for India in these data (Supplementary Tables S27 and S165).

## Discussion

Although previous studies empirically linked various childhood factors and smoking in adolescence and adulthood, they were conducted mostly in Western countries in North America, Europe, and Oceania[4–8,10–13,15–17,25,26,53–61] and rarely synthesized the effects of those factors on adolescent or adult smoking in culturally and geographically diverse countries despite the availability of various sources of cross-national data [https://www.healthdata.org/research-analysis/gbd]. In addition, they relied almost exclusively on a binary measure of smoking, rarely employing a continuous measure, while the quantity of daily consumption is a key predictor of tobacco-associated disease risk[1]. To address these oversights, using nationally representative data from 22 countries that participated in the Global Flourishing Study, we meta-analyzed country-specific effects of childhood predictors and demographic variables on both binary and continuous measures of daily smoking in adulthood.

Our childhood factors mostly had anticipated associations with daily cigarette smoking in adulthood, consistent with prior research (Research Question #1). For example, maternal and paternal relationship quality (both total and smoker samples) and religious service attendance (smoker sample) predicted a lower likelihood of adult daily smoking, whereas being raised by a divorced parent (total sample), physically or sexually abused (both samples), and an outsider in the family when growing up (both samples) predicted a higher likelihood. We also found that poor health growing up and a proxy of childhood immigration status (being born in another country), both understudied by previous researchers, predicted a higher and lower likelihood of daily smoking in adulthood, respectively. The direction and strength of these associations were generally consistent between the alternative measures of daily smoking (binary vs. continuous) and samples analyzed (total vs. smoker) and moderately robust against potential unmeasured confounding with a few exceptions (Research Question #3). Furthermore, the strength and direction of associations varied across countries due likely to the influence of diverse—whether sociocultural, economic, or health—contexts that characterize each nation (Research Question #2), though it was not always immediately clear.

In the total sample analysis, for example, since attending religious services at least once a week in childhood had preventive effects on daily smoking below an RR of 0.90 in 32% of GFS countries and causative effects above an RR of 1.10 in 14% of the countries (Table 3), we compared the latter countries (Nigeria, Sweden, and Tanzania) with the former (Australia, Kenya, Mexico, the Philippines, Poland, South Africa, and the U.S.) but found no particular difference in religious context, as both groups of countries had the same primary religion, Christianity, except for Nigeria that had two major religions, 49% Islam as well as 51% Christianity (see Supplementary Tables S9, S42, S46, S50, S54, S58, S62, S70, S78, and S86). In the smoker sample analysis, however, we found that childhood religious attendance of any frequency had a relatively large, salutary effect on the quantity of daily cigarette consumption in adulthood in 95–100% of those religiously diverse countries. This finding may indicate that the health risk-reducing effect of childhood religious service attendance on daily cigarette consumption in adulthood is independent of religious context, as individuals who attended religious services in childhood and became adult smokers are likely to consume less cigarettes daily compared to those who never attended religious services growing up, regardless of which religious context they were in.

Despite this finding, it is still worth studying whether religion functions as a context for smoking differently across countries and that some other factors besides religion (e.g., smoking culture or smoking used as a substitute for drinking, which is strictly prohibited in Muslim countries) affect the health-risk behavior given that all those religious traditions discourage health-risk behaviors, including smoking. For example, one potentially fruitful topic for future research is subgroup analysis of the quantity of daily cigarette smoking to compare countries that are different in the primary

religion (e.g., predominantly Christian vs. Muslim countries) and smoking culture (for which smoking prevalence could be used as a proxy measure given that it is not highly correlated with smoking intensity, that is, quantity[62]) to see whether the cultural factor is a significant confounder for the relationship between religion and smoking. Alternatively, researchers could compare regional subgroups of individuals equally involved in the same religion (e.g., Christians attending religious services at least once a week in African, Asian, and American regions)[19].

Also, consistent with previous studies on gender differences in smoking[20,23,24], above-average negative effects of being female on the binary measure of daily smoking (compared to male) were found mostly in countries with traditional gender roles and socialization (Egypt, Tanzania, Indonesia, India, Kenya, Nigeria, the Philippines, and South Africa; Supplementary Fig. S26). Generally, the same pattern was found for the continuous measure in the smoker sample (Türkiye, Egypt, Israel, the Philippines, Indonesia, Argentina, Poland, Spain, and Japan; Supplementary fig. S80). The smoker-sample analysis also revealed that being female had a positive effect on daily smoking in Kenya (1.98), India (2.56), Nigeria (3.57), and Tanzania (6.78): that is, female smokers consumed, on average, about two or more cigarettes per day than male smokers (Supplementary Fig. S80). This finding, however, should be interpreted with caution given that these countries had low prevalence of daily smoking—4% (Tanzania and Nigeria), 5% (Kenya), and 8% (India)—and that their female effects were all negative in the binary analysis (Supplementary Fig. S26). That is, it seems unlikely that women smoke more cigarettes daily than men in a developing country where a relatively small percentage of population are daily smokers with female prevalence of smoking being lower than male prevalence. Furthermore, the gender effect had large 95% CI including 0 in all four countries: Kenya (1.98, [−2.23, 6.18]), India (2.56, [−3.98, 9.09]), Nigeria (3.57, [−3.21, 10.36]), and Tanzania (6.78, [−5.22, 18.78]), which may indicate that their smoker samples were biased.

Although random effects meta-analysis showed that "good" relationships with mother and father in childhood both had preventive effects on the prevalence of daily smoking in adulthood as expected[5–8], a closer examination revealed that a strong positive effect of maternal relationship quality was found in a predominantly Muslim country, Egypt, while that of paternal relationship quality was negative (see Supplementary figs. S1 and S2). Together with the positive effect of parental divorce in childhood on adult smoking in that country (Supplementary Fig. S3), we speculate that the positive effect of maternal relationship quality on adult smoking may partly reflect the influence of a mother, whether married or divorced, who smoked to cope with a strained relationship with the father who may have been under-involved in marriage and parenting, or post-marriage stress due to social disapproval, stigmatization, as well as the financial challenges related to divorce, as found in previous studies conducted in Muslim countries including Egypt[63–68]. Given that mothers tend to be the primary caregiver, it is possible that children may learn smoking behavior from their mother in early years of life, but these speculations extend beyond the present data.

Meta-analytic effects of childhood predictors were generally consistent between the continuous and binary measures of daily smoking and between the total and smoker samples, while prior research rarely made either comparison. This finding confirms previous studies of childhood predictors using a binary measure of smoking, but it does not mean that studying the quantity of smoking is unnecessary, as our analysis did not include some of the key predictors, such as modeling of smoking by others and mental health in childhood[3,17,53,54,56,60,69]. These and other omitted predictors—such as genetic/biological (e.g., intelligence, maternal smoking during pregnancy, and puberty)[17,59,61] and psychological factors (e.g., personality and self-control)[9,15,53,55,57,58]—need to be examined to see whether they similarly predict alternative measures of smoking and between total and smoker samples. Studying childhood predictors of daily cigarette consumption in adulthood is particularly important for smokers, as number of cigarettes smoked daily is a key predictor of tobacco-associated disease risk[1].

Finally, we acknowledge key limitations of our study. First, although self-report is a common method to measure smoking in population-based studies, like the current study, prior research using biomarkers (e.g., cotinine) confirmed the method's underestimation bias due to socially desirable reporting[70–72], while this bias does not necessarily negate the validity of self-reported smoking[4,73]. For example, participants in countries with aggressive anti-tobacco policies and legislations (e.g., Brazil, Spain, and Türkiye)[74] might have been more likely to underreport smoking than those in countries without such measures. Second, since we used retrospective data on childhood factors to predict smoking in adulthood, there may be potential recall bias as well as response error. However, for recall bias to completely explain away the observed associations would require that the effect of adult smoking on biasing retrospective assessments of the childhood predictors would essentially have to be at least as strong as the observed associations themselves[75], and some of these were moderately substantial. Third, the use of a retrospective recall of childhood characteristics to create a synthetic longitudinal design also limits not only the generalizability of the results due to recall bias but also the ability to mitigate omitted confounding. In companion methods papers[19,33], the development of a robust set of childhood predictors are discussed, and issues we encountered in the implementation of these methods are documented. The development of a robust set of childhood predictors was how we aimed to mitigate the influence of omitted confounding in this synthetic longitudinal design. Still, future waves of panel data will provide opportunities to mitigate confounding more confidently. The sensitivity analyses by reporting E-values do not mitigate omitted confounding but do provide a metric, on the risk-ratio scale, on which to evaluate the strength such a confounder would have to have to move the observed effects to null effects.

Fourth, in the analyses on continuous cigarettes smoking but restricted to the smoking sample, the restriction/conditioning on smokers, an outcome which can be affected by childhood experience, can itself introduce spurious associations and so these results on the smokers-only sample need to be interpreted cautiously. In addition, the sample size of daily cigarette smokers was smaller in some countries (e.g., Nigeria and Australia) than others, so their country-specific findings may be of relatively limited interpretation. Lastly, while the Global Flourishing Study took necessary steps (including cognitive and pretest interviews) to ensure that survey items would be understood similarly across countries[31,38,76], participants' interpretation of some items might have varied across different cultures. For example, "good" relationship with parent might have meant different relational quality to survey respondents (e.g., conflict-free vs. affectionate relationship), depending on their country's culture, and thus may have been understood differently when they answered the items about their relationship with mother (e.g., "good" = affectionate) and father (e.g., "good" = conflict-free).

In conclusion, based on new data from 22 countries, this cross-national study contributes to global research on tobacco use, as we meta-analyzed country-specific effects of childhood predictors on daily cigarette smoking in adulthood, measuring the outcome as continuous as well as binary variable and predicting the quantity of daily cigarette consumption in a smoker sample. Results from synthesizing those effects tend to be similar between the total and smoker samples. While the results are generally consistent with previous findings based on a binary measure of smoking, future research should examine the continuous measure of daily smoking for childhood predictors omitted in the present study.

## Data availability

The data (https://doi.org/10.17605/OSF.IO/3JTZ8) that support the findings of this article are publicly available on the Open Science Framework (https://www.cos.io/gfs-access-data)[35].

## Code availability

All code (https://doi.org/10.17605/Osf.Io/Vbype) to reproduce analyses are openly available in an online repository[32].

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

## Acknowledgements

The GFS was supported by funding from the John Templeton Foundation (grant #61665), Templeton Religion Trust (#1308), Templeton World Charity Foundation (#0605), Well-Being for Planet Earth Foundation, Fetzer Institute (#4354), Well Being Trust, Paul L. Foster Family Foundation, and the David and Carol Myers Foundation.

## Author contributions

S.J.J. conducted the literature review, performed the data analysis, interpreted the results, and drafted the full manuscript. S.J.J. and P.A.D.L.R. conducted literature searches and collaborated on early versions of the manuscript. R.N.P, M.B., and S.J.J. developed code for data analysis. R.N.P. contributed to the interpretation of the results. T.J.V. and B.R.J. contributed to the study concept and design, coordinated data collection, participated in survey design, coordinated the development of code for data analysis, and contributed to the interpretation of the results. All authors contributed to revisions of the manuscript. All authors read and approved the final manuscript.

## Competing interests

The authors declare the following competing interests: T.J.V. reports consulting fees from Gloo Inc., along with shared revenue received by Harvard University in its license agreement with Gloo according to the University IP policy. All other authors declare no competing interests.
