## [Transparent Peer Review file · Communications Medicine]

A cross-national analysis of childhood predictors of daily smoking in adulthood

Corresponding Author: Dr Sung Joon Jang

Version 0:

Reviewer comments:

Reviewer #1

(Remarks to the Author)

This study is to examine whether childhood factors predict adult smoking using data from a large global survey covering 22 countries. Using the random effects meta-analyses approach, the authors presented comprehensive analysis results.

This study used a large cross-national data – Global Flourishing Survey (Wave 1) – to estimate the associations of childhood predictors with smoking in adulthood. The authors stated that the GFS data provide childhood measures and current smoking measures (binary and continuous) that were not available in any other global databases in the past. Little is known as to what specific childhood factors are “critical” to reduce adult smoking. This study therefore would contribute to the field of population health, if it provides empirical evidence that some “universal” childhood factors predict smoking in adulthood. Such information can help estimate health burden related to smoking and inform global policy makers to take actions to reduce the number of smokers worldwide.

1. Strong theoretical, conceptual models are needed to lead readers to the hypotheses. Although the introduction section described some previous research listing a number of individual and social factors predicting adult smoking behaviors, many factors are individually associated with adult smoking, and there is no clear conceptual model describing a pathway of how everything in this list work together leading individuals to smoking in adulthood. Currently, this section reads like a random list of available variables to explore relationships with smoking outcome variables.
2. The hypotheses should have been more specific to clearly indicate which factors are associated with adult smoking and how. For example, Hypothesis 1 listed all the available variables from the data set and stated “will show meaning associations...” How do you define “meaningful?” Any specified level of strengths or directions compared to previous studies, or statistical significance? Hypothesis 2 is also stated vaguely (“will vary by country.”). It should be written to test specific relationships and directionality, based on data and previous studies providing expected relationships by country/region. Hypothesis 3 does not read like a hypothesis: “The observed associations between the 13 childhood predictors and an individual's daily smoking in adulthood will be robust against potential unmeasured confounding.” This is a research question focused on methodology, not a primary hypothesis at best.
3. The first sentence of status of tobacco use is a blanket statement. For the past 10 years tobacco use among young and old people in the U.S. and in the world has changed quite a bit. In the U.S. smoking has been declined over the last decade, especially youth smoking (see <https://www.cdc.gov/tobacco/php/data-statistics/trends-in-tobacco-use-among-youth.html>; Nkosi L, Odani S, Agaku IT. 20-Year Trends in Tobacco Sales and Self-Reported Tobacco Use in the United States, 2000-2020. *Prev Chronic Dis.* 2022 Jul 28;19:E45. doi: 10.5888/pcd19.210435. PMID: 35900882; PMCID: PMC9336190.). Smoking is still a big problem, but I would not characterize that tobacco use is a ‘pediatric epidemic’ and cite the 2012 U.S. Surgent General Report as a reference.
4. Methods: Please provide distributional data for the main outcome - smoking data for binary and continuous measures for the total and smokers samples. Table 1 lists only predictors. For the primary analyses, Table 2 presents regression results with continuous smoking (quantity of daily cigarette consumption) using the total sample. Smokers and non-smokers are clearly different populations (even if some non-smokers might have been previous smokers), however. What was the rationale to combine both populations in one sample to estimate the effects of the predictors on the daily cigarette consumption? Also, please provide statistical justifications for combining both smokers and non-smokers in running linear regressions. It needs to be explained how the meta-analyses regressions treat these data with skewed distributions.
5. Results and Discussion: More in-depth interpretation of the results is desirable. There are binary and quantity smoking outcome analyses with several predictors. So the following findings need to be reported and discussed: (1) How the

estimated effects were translated into the outcome variables (e.g., what does it mean by the estimated parameter of -0.12 for reporting "poor" health growing up presented in Table 2)?; (2) What childhood predictors were relatively more important?; (3) How the impacts of these childhood predictors are compared to the impacts of age, sex, and race (i.e., non-childhood predictors)?"

6. Results and Discussion: I wish for more in-depth discussions from the sensitivity analysis, e.g., strategies to mitigate omitted confounders?

7. Discussion: What subgroup analyses could be useful to delineate the relationships between predictors and smoking, after the meta-analyses using data from 22 countries that have diverse contextual and cultural variations for smoking (e.g., Australia vs. Türkiye or regional sub-groups)?

Reviewer #2

(Remarks to the Author)

This cross-national study of 22 countries seeks to establish the association between childhood characteristics and daily cigarette smoking in adulthood. The author use the novel Global Flourishing Study dataset of 200,000 adults to achieve this goal, and report a variety of self-reported childhood characteristics were associated with self-reported daily cigarette smoking in adulthood.

The authors claim at several points that 'cross-national studies of smoking are scarce' which is not true. The Demographic and Health Survey series, Multiple Indicator Cluster Surveys, WHO STEPwise studies, etc. all achieve this goal using standardized approaches, spanning hundreds of national surveys over decades. The authors can find a more comprehensive list of data sources by examining the Global Burden of Diseases Study report on smoking (with survey details found in the supplement of the most recent report). The manuscript would be strengthened by more precisely pointing out the gap in the literature, which is that types of childhood characteristics the authors are interested in are not captured in these surveys, which makes identifying these specific potential predictors of adult smoking difficult and under-studied at scale.

Limitations of this study include its retrospective and self-reported nature, which would be expected to lead to some bias, as the authors adequately note in the discussion. The authors don't comment on the fact that the sample size of daily cigarette smokers broken down by country is actually quite small, and therefore country-specific findings may be of limited interpretation.

Given the longitudinal nature of this study, as well as its sensitive questions about childhood (eg, sexual abuse), it must have required collecting fairly comprehensive personal information. I was therefore surprised this study was exempted from IRB review. Could the authors please comment on this?

Was information about former smoking collected? Although continued daily smoking is of course more strongly associated with mortality risk than former smoking, this information on predictors of cessation is at least as valuable given the benefits of quitting throughout adulthood.

The authors state that missing data on all variables were imputed by MICE. Is information about this available somewhere? Eg, was missingness rare or common, and did it vary by country or variable? This should probably be presented in detail in the supplement.

I found it somewhat odd that only the total and smoker sample were reported, as it would be more traditional and informative to present smoker, non-smoker, and then total results separately (though I recognize this can be deduced, and so would defer to the authors' preference).

More detailed reminders in the footnotes of each table clarifying how to interpret the presented findings would make the manuscript more readable. For example, scanning across e-values (an uncommonly reported metric) and e-value limits for continuous vs binary results in the total vs smoker sample is not at all intuitive.

Version 1:

Reviewer comments:

Reviewer #1

(Remarks to the Author)

The authors have addressed most of my comments in the revised manuscript. I appreciate the authors' effort. I believe this revised manuscript clearly communicates the research questions, results, limitations, and discussion.

I have one remaining concern, however, regarding my question in comment # 4 (copied below). Table 2 shows the results with the total sample (N = 202,898) only for the continuous daily cigarette measure. Given the small smoker sample (N = 38,290), I still do not think it is appropriate to run a model combining a large number of zeros and positive numbers. Usually, this type of analysis uses a zero-inflated model or two parts model. I agree that regression models are robust, so some

degree of violation of normality assumption and heteroscedasticity is acceptable. If you compare the estimates of using only smokers and compare the results between total sample and smokers only, do they come up as similar?

4. Methods: Please provide distributional data for the main outcome - smoking data for binary and continuous measures for the total and smokers samples. Table 1 lists only predictors. For the primary analyses, Table 2 presents regression results with continuous smoking (quantity of daily cigarette consumption) using the total sample. Smokers and non-smokers are clearly different populations (even if some non-smokers might have been previous smokers), however. What was the rationale to combine both populations in one sample to estimate the effects of the predictors on the daily cigarette consumption? Also, please provide statistical justifications for combining both smokers and non-smokers in running linear regressions. It needs to be explained how the meta-analyses regressions treat these data with skewed distributions.

A suggestion:

I gained a better understanding on the background for the indicator selections in the GFS survey and the explanation of "not having a specific conceptual model" for this manuscript. Still, the GFS is probably guided by the concept of "human flourishing." I suggest that the authors include a short description of the GFS's leading conceptual framework, which links to the rationale behind selection of GFS survey question items. That framework might help readers better understand the aim of this study as exploratory work to investigate multiple pathways of individual's well-being in a larger context.

Reviewer #2

(Remarks to the Author)

Thank you for addressing the concerns raised. I think this report will be a meaningful contribution to the field of smoking and its associated predictors in diverse settings globally.

Version 2:

Reviewer comments:

Reviewer #1

(Remarks to the Author)

My concern regarding the statistical approach for Table 2 remains. As I mentioned in the previous review, Table 2 shows the results the continuous daily cigarette measure with the total sample (N = 202,898). Given the small smoker sample (N = 38,290), I still believe it is inappropriate to run a model combining a large number of zeros (81% - no daily cigarettes) with positive values (19%).

When comparing the results in Table 2 with those in Table 5, I notice differences. I do not completely agree with your explanation regarding potential sources of differences in some predictors. I believe the model used for Table 2 is mis-specified, leading to biased estimates. I suggest that the models should include a binary outcome using the total sample, and then a continuous outcome using the smoker sample.

Version 3:

Reviewer comments:

Reviewer #1

(Remarks to the Author)

I appreciate the scope and scale of this study, particularly the extensive data collected across 22 countries. I recognize the value of such a large, multi-country dataset and the effort involved in the analyses. However, I am having difficulty fully understanding or endorsing the authors' interpretations of the results.

With the revised approach and analysis results, the authors presented the estimated effects of the same set of predictors on two outcomes: (1) the likelihood of being a daily smoker and (2) the number of cigarettes smoked daily. However, I am having difficulty understanding how the authors interpreted the estimated effects, particularly for certain predictors. In several cases, their interpretations appear to be overstated or not well-supported by the reported estimates.

Specifically, I am trying to understand the interpretation of relative risks (RRs) presented in Table 2. Take, for example, the predictor "religious service attendance at age 12." The reported RRs for the three comparison categories relative to "never" attending religious services were approximately 1.0 (ranging from 0.96 to 1.05 – weak effects), suggesting little to no association with the outcome. The proportion of effects below and above the defined threshold were 32% and 14%, respectively – interpreted as roughly 7 countries showing RRs below the threshold and 3 above, out of 22 countries. The remaining 12 countries appear to have RRs close to 1.00, indicating no strong association with religious service attendance at age 12. Is this interpretation correct?

If so, I am unsure how the authors concluded that religious service attendance was a protective factor against daily smoking.

This interpretation seems inconsistent with the reported RRs and the proportions of effects crossing a meaningful threshold.

Age 12 religious service attendance (Copied from Table 2)

(Ref: Never) Estimated proportion of effects by threshold
At least 1/week 0.96 (0.90,1.03) 0.32 < 0.90 > 1.10
1-3/month 1.05 (0.95,1.16) 0.14 0.36 75.7
< 1/month 1.00 (0.95,1.05) 0.05 0.05 30.5

I encountered similar issues with the authors' interpretation of other predictors. For example, in the models of binary smoking, the reported effects for socioeconomic status (SES) and health were not consistently strong or unidirectional. In the models for daily cigarette consumption, the effects of parental marital status, some SES categories, health status, outsider status, and immigration background were also mixed or weak in magnitude. Overall, the direction and strength of the estimated effects for these predictors do not clearly support the conclusions drawn in the Abstract's Results section. As such, I do not fully agree with the authors' interpretations.

Point-by-point response to reviewers' comments

Reviewer #1 (Remarks to the Author):

This study is to examine whether childhood factors predict adult smoking using data from a large global survey covering 22 countries. Using the random effects meta-analyses approach, the authors presented comprehensive analysis results.

This study used a large cross-national data – Global Flourishing Survey (Wave 1) – to estimate the associations of childhood predictors with smoking in adulthood. The authors stated that the GFS data provide childhood measures and current smoking measures (binary and continuous) that were not available in any other global databases in the past. Little is known as to what specific childhood factors are “critical” to reduce adult smoking. This study therefore would contribute to the field of population health, if it provides empirical evidence that some “universal” childhood factors predict smoking in adulthood. Such information can help estimate health burden related to smoking and inform global policy makers to take actions to reduce the number of smokers worldwide.

1. Strong theoretical, conceptual models are needed to lead readers to the hypotheses. Although the introduction section described some previous research listing a number of individual and social factors predicting adult smoking behaviors, many factors are individually associated with adult smoking, and there is no clear conceptual model describing a pathway of how everything in this list work together leading individuals to smoking in adulthood. Currently, this section reads like a random list of available variables to explore relationships with smoking outcome variables.

- As explained in our next response (#2) below, our research was pre-registered (with the Center for Open Science) as an exploratory rather than confirmatory study. Thus, the term “hypothesis,” used in the paper, was a misnomer and potentially misleading despite our use of the qualifier, “working,” to indicate the exploratory nature of the study. To address this confusion, we replaced “working hypotheses” with “research questions,” which were also pre-registered (p. 4).
- The reviewer’s comment also offers us an opportunity to clarify the selection of childhood predictors. All those variables were selected based on theories predicting flourishing (e.g., health) and its related variables (e.g., smoking), as a part of the design of the Global Flourishing Study (GFS). A total of 17 retrospective recall items about childhood characteristics and experiences were included in the GFS intake survey, and the choice of indicators for use in the survey was a multi-phase process that is described in detail elsewhere (Lomas et al. The development of the Global Flourishing Study questionnaire: Charting the evolution of a new 109-item inventory of human flourishing. *BMC Global & Public Health* [in press]) (p. 3).
- Our study is a part of a coordinated collection of studies evaluating multiple aspects of flourishing and using a harmonized set of predictors (p. 3). Since all these GFS studies use the same prediction model to enhance comparability, we did not apply a theory-based approach (e.g., testing a conceptual model that describes a pathway linking childhood factors to adult smoking), while using theory-based childhood predictors of smoking in adulthood.
- Although we initially pre-registered to use all the 17 items as childhood predictors, the final set of childhood predictors was reduced to 11 after pre-testing due to issues of multicollinearity. The selected items were chosen based largely on theory and in consultation with collaborators at Gallup (who conducted the sampling) on which items were appropriate to keep. For example, the childhood predictors of child’s, mother’s, and father’s religious service attendance were highly collinear leading to the need to select only one such variable. The final set of variables was carefully selected to be as broadly meaningful as possible of childhood determinants of any

measure of human flourishing. Still, multicollinearity issues led to some conceptually distinct variables being omitted or modified, such as maternal/paternal relationship quality variables being collapsed to binary predictors (p. 3). These changes were applied to the entire collection of GFS studies on childhood predictors of flourishing and its related variables.

2. The hypotheses should have been more specific to clearly indicate which factors are associated with adult smoking and how. For example, Hypothesis 1 listed all the available variables from the data set and stated “will show meaning associations...” How do you define “meaningful?” Any specified level of strengths or directions compared to previous studies, or statistical significance? Hypothesis 2 is also stated vaguely (“will vary by country.”). It should be written to test specific relationships and directionality, based on data and previous studies providing expected relationships by country/region. Hypothesis 3 does not read like a hypothesis: “The observed associations between the 13 childhood predictors and an individual's daily smoking in adulthood will be robust against potential unmeasured confounding.” This is a research question focused on methodology, not a primary hypothesis at best.

- We appreciate this point being raised. While we understand the Reviewer’s concern, we respectfully note that our paper was pre-registered as exploratory (p. 4). Again, that is why we used the qualifier, “working,” when we introduced our hypotheses. Adding specific hypotheses at this stage is discouraged by the pre-registration guidelines (i.e. because it violates the principal of avoiding ‘hypothesizing after the results are known’).
- In order to address the Reviewer’s concern, however, agreeing that our “working hypotheses” are better referred to as “research questions” than hypotheses, we have replaced our hypotheses with pre-registered research questions to highlight the exploratory nature of our analysis (p. 4), while still specifying anticipated associations between childhood predictors and adult smoking based on prior research and general expectations about how those associations will vary by country and be robust against potential unmeasured confounding (pp. 4-6).

3. The first sentence of status of tobacco use is a blanket statement. For the past 10 years tobacco use among young and old people in the U.S. and in the world has changed quite a bit. In the U.S. smoking has been declined over the last decade, especially youth smoking (see <https://www.cdc.gov/tobacco/php/data-statistics/trends-in-tobacco-use-among-youth.html>; Nkosi L, Odani S, Agaku IT. 20-Year Trends in Tobacco Sales and Self-Reported Tobacco Use in the United States, 2000-2020. *Prev Chronic Dis.* 2022 Jul 28;19:E45. doi: 10.5888/pcd19.210435. PMID: 35900882; PMCID: PMC9336190.). Smoking is still a big problem, but I would not characterize that tobacco use is a ‘pediatric epidemic’ and cite the 2012 U.S. Surgeon General Report as a reference.

- We appreciate the Reviewer pointing out this concern. In response, we revised the first sentence to state the decreasing trends of tobacco use, including cigarette smoking, globally and in the U.S., citing studies by Reitsma et al. as well as Nkosi et al. (p. 1).

4. Methods: Please provide distributional data for the main outcome - smoking data for binary and continuous measures for the total and smokers samples. Table 1 lists only predictors. For the primary analyses, Table 2 presents regression results with continuous smoking (quantity of daily cigarette consumption) using the total sample. Smokers and non-smokers are clearly different populations (even if some non-smokers might have been previous smokers), however. What was the rationale to combine both populations in one sample to estimate the effects of the predictors on the daily cigarette consumption? Also, please provide statistical justifications for combining both smokers and non-smokers in running linear regressions. It needs to be explained how the meta-analyses regressions treat these data with skewed distributions.

- Given our space constraints, we did not include in Table 1 the frequency distribution of the continuous measure of daily smoking, as it ranges from 0 to 97. In response to the reviewer’s suggestion, however, we report summary statistics of binary (prevalence rate) as well as the continuous measure of daily smoking (mean and standard deviation) for the total sample and by country in a supplemental table as alternative distributional data for the main outcome (p. 11).
- We first present regression results with continuous and binary measures of daily smoking for the total sample that combines current smokers and non-smokers, since our study relies upon a population-based sample. We then report results with the continuous measure for smokers to show whether the effects of childhood and demographic factors on daily smoking differed in the smoker sample compared to the total sample.
- All regression analyses were conducted within the country, and then we pooled the estimated regression coefficients. When treated as continuous, the skewed nature of these data can influence the estimated standard errors in normal linear regression; however, the complex survey-adjusted robust standard errors are generally less influenced by non-normality. Additionally, to the best of our understanding, the point estimates of the regression coefficients are generally not influenced by non-normality but can sometimes be attenuated towards zero, making our results more conservative. No meta-regressions were conducted.

5. Results and Discussion: More in-depth interpretation of the results is desirable. There are binary and quantity smoking outcome analyses with several predictors. So the following findings need to be reported and discussed: (1) How the estimated effects were translated into the outcome variables (e.g., what does it mean by the estimated parameter of -0.12 for reporting “poor” health growing up presented in Table 2)?; (2) What childhood predictors were relatively more important?; (3) How the impacts of these childhood predictors are compared to the impacts of age, sex, and race (i.e., non-childhood predictors)?”

- As suggested, more in-depth interpretation of the results is provided by (1) translating meta-analyzed effects into the continuous (p. 14) and binary measures of daily smoking (p. 21), using the example of maternal and paternal relationships quality results, (2) discussing which childhood predictors were relatively more important (p. 25), and (3) comparing the relative importance between childhood predictors and demographic variables (p. 25).

6. Results and Discussion: I wish for more in-depth discussions from the sensitivity analysis, e.g., strategies to mitigate omitted confounders?

- As suggested, we added a discussion in relation to an additional limitation of using retrospective data on childhood predictors (p. 31).

7. Discussion: What subgroup analyses could be useful to delineate the relationships between predictors and smoking, after the meta-analyses using data from 22 countries that have diverse contextual and cultural variations for smoking (e.g., Australia vs. Türkiye or regional sub-groups)?

- In response to the Reviewer’s suggestion, we added a discussion of potentially fruitful topics for future research on subgroup analysis (p. 28).

Reviewer #2 (Remarks to the Author):

This cross-national study of 22 countries seeks to establish the association between childhood characteristics and daily cigarette smoking in adulthood. The author use the novel Global Flourishing Study dataset of 200,000 adults to achieve this goal, and report a variety of self-reported childhood

characteristics were associated with self-reported daily cigarette smoking in adulthood.

The authors claim at several points that 'cross-national studies of smoking are scarce' which is not true. The Demographic and Health Survey series, Multiple Indicator Cluster Surveys, WHO STEPwise studies, etc. all achieve this goal using standardized approaches, spanning hundreds of national surveys over decades. The authors can find a more comprehensive list of data sources by examining the Global Burden of Diseases Study report on smoking (with survey details found in the supplement of the most recent report). The manuscript would be strengthened by more precisely pointing out the gap in the literature, which is that types of childhood characteristics the authors are interested in are not captured in these surveys, which makes identifying these specific potential predictors of adult smoking difficult and understudied at scale.

- We appreciate the Reviewer pointing this out and agree that the statement is not true. We should have said: Cross-national studies synthesizing (i.e., meta-analyzing) the effects of childhood predictors on smoking are scarce. So, we revised accordingly in the abstract as well as text (ps. 1, 3, 26, 32). We also cited the Global Burden of Diseases website to indicate that various cross-national data sources are available (ps. 3, 26).

Limitations of this study include its retrospective and self-reported nature, which would be expected to lead to some bias, as the authors adequately note in the discussion. The authors don't comment on the fact that the sample size of daily cigarette smokers broken down by country is actually quite small, and therefore country-specific findings may be of limited interpretation.

- As suggested, we added the limitation associated with relatively small sample size of daily cigarette smokers in the discussion (p. 31).

Given the longitudinal nature of this study, as well as its sensitive questions about childhood (eg, sexual abuse), it must have required collecting fairly comprehensive personal information. I was therefore surprised this study was exempted from IRB review. Could the authors please comment on this?

- First, we revised sentences regarding IRB review to provide detailed information about ethical approval granted by the institutional review boards at the lead author's university and Gallup, citing a paper on GFS survey design (pp. 6-7).
- Second, while we appreciate the reviewer's concern about the sensitive nature of the question, our study was exempt from the university IRB review because the data on physical or sexual abuse were collected retrospectively by asking *adult* respondents regarding their *childhood* experiences.
- Third, although our study collected "fairly comprehensive personal information," the cited survey design paper (Padgett et al. Survey sampling design in wave 1 of the Global Flourishing Study. *European Journal of Epidemiology* [in press]; p. 7) describes how we protected the privacy of study participants as follows:
 - "Respondent confidentiality is maintained by giving a pseudo-random identification number that is kept separate from the data that researchers can access. A public use dataset ... excludes individual identifying information such as location (latitude and longitude are rounded to the nearest degree) and language of assessment. Individual researchers with appropriate institutional review board approval may be able to access a restricted use dataset that contain these more sensitive types of data."

Was information about former smoking collected? Although continued daily smoking is of course more strongly associated with mortality risk than former smoking, this information on predictors of cessation is at least as valuable given the benefits of quitting throughout adulthood.

- Unfortunately, information about former smoking was not collected, while we agree that such information would have enabled us to examine predictors of cessation as well as current smoking. The multi-purpose nature of the GFS questionnaire that contains items tapping various domains of life enables us to study distinct aspects of human flourishing holistically, but it allowed little space for items needed to study these concepts with real depth.

The authors state that missing data on all variables were imputed by MICE. Is information about this available somewhere? E.g., was missingness rare or common, and did it vary by country or variable? This should probably be presented in detail in the supplement.

- Missingness was rare in the combined total and smoker samples (less than 5% on all variables) with one minor exception (5.4% on parental marital status in the smoker sample) (ps. 11, 14).
- The information about missingness for each country is available in our online supplement under Tables S1a-S22a (total sample) and Tables S49a-S70a (smoker sample), where the summary statistics (including the number and percent missing) are reported for each variable (p. 14).
- Missingness did vary by country. However, it was generally rare with a few variables (e.g., parental marital status) having relatively high rates of missing in some countries, but, on average, missingness was not a concern (p. 14).

I found it somewhat odd that only the total and smoker sample were reported, as it would be more traditional and informative to present smoker, non-smoker, and then total results separately (though I recognize this can be deduced, and so would defer to the authors' preference).

- Because our study was population-based research, we first presented results for the total sample (Tables 2 and 4), followed by those for current smokers (Table 5) to show whether the same childhood predictors behaved differently in the smoker sample, while non-smokers are not relevant to our analysis of daily cigarette consumption.
- Besides, as acknowledged above, our item about daily smoking does not allow us to accurately identify people as non-smokers, since respondents who reported smoking less than 1 cigarette per day (coded as 0) are included in the same group as those who never smoked in life. Because of this limitation as well as the population-based nature of the study, we decided to focus on the smokers.

More detailed reminders in the footnotes of each table clarifying how to interpret the presented findings would make the manuscript more readable. For example, scanning across e-values (an uncommonly reported metric) and e-value limits for continuous vs binary results in the total vs smoker sample is not at all intuitive.

- While Tables 2, 4, and 5 had a note on the meaning of key statistics, Table 3 did not. So, as suggested, we added a note for the sensitivity analyses (p. 20).

Point-by-point response to reviewers' comments

Reviewer #1 (Remarks to the Author):

The authors have addressed most of my comments in the revised manuscript. I appreciate the authors' effort. I believe this revised manuscript clearly communicates the research questions, results, limitations, and discussion.

I have one remaining concern, however, regarding my question in comment # 4 (copied below). Table 2 shows the results with the total sample ($N = 202,898$) only for the continuous daily cigarette measure. Given the small smoker sample ($N = 38,290$), I still do not think it is appropriate to run a model combining a large number of zeros and positive numbers. Usually, this type of analysis uses a zero-inflated model or two parts model. I agree that regression models are robust, so some degree of violation of normality assumption and heteroscedasticity is acceptable. If you compare the estimates of using only smokers and compare the results between total sample and smokers only, do they come up as similar?

- We appreciate the concern of accounting for potential misspecification in the regression model and how this might influence our results. We point the reviewer to Tables 2 and 5 for the results for the complete sample ($N = 202,898$) and daily smoker only sample ($N = 38,290$), respectively. In our previous version of this paper, we used these results to illustrate that the results were generally similar with respect to the effects of childhood predictors on adult daily cigarette consumption with some exceptions (see pages 23 & 25 of this version of the manuscript). In the few instances where the results were different, we discussed potential sources of these observed differences in childhood predictor effects (i.e., smaller variance of childhood abuse in the daily smoker sample, cross-national variation in the effect of being raised by divorced parent, and greater influence of religious service attendance in childhood).
- The results for the daily smoker sample would be similar to the non-zero component of a zero-inflated model and the analyses reported in Table 4, predicting smoking prevalence, would be the inverse of the zero-inflated portion. Zero-inflated models predict the 0's while the results presented on prevalence predict the 1's. So, while we did not directly estimate a zero-inflated model, the results from Tables 4 and 5 combine to what we would get from a zero-inflated model, if only approximately. As part of the coordinated analytic strategy of this paper with other researchers, a computational limitation was our reliance on the survey package in R, which, to the best of our knowledge, cannot be used to estimate zero-inflated models with design corrected standard errors. So, a piece-wise approach was our second-best option.

4. Methods: Please provide distributional data for the main outcome - smoking data for binary and continuous measures for the total and smokers samples. Table 1 lists only predictors. For the primary analyses, Table 2 presents regression results with continuous smoking (quantity of daily cigarette consumption) using the total sample. Smokers and non-smokers are clearly different populations (even if some non-smokers might have been previous smokers), however. What was the rationale to combine both populations in one sample to estimate the effects of the predictors on the daily cigarette consumption? Also, please provide statistical justifications for combining both smokers and non-smokers in running linear regressions. It needs to be explained how the meta-analyses regressions treat these data with skewed distributions.

A suggestion:

I gained a better understanding on the background for the indicator selections in the GFS survey and the explanation of “not having a specific conceptual model” for this manuscript. Still, the GFS is probably guided by the concept of “human flourishing.” I suggest that the authors include a short description of the GFS’s leading conceptual framework, which links to the rationale behind selection of GFS survey question items. That framework might help readers better understand the aim of this study as exploratory work to investigate multiple pathways of individual’s well-being in a larger context.

- In response to the reviewer’s suggestion, we added the GFS’s conceptualization of human flourishing as well as a brief description of how GFS survey items, particularly, childhood predictors of flourishing in adulthood were selected (pp. 3-4).

Reviewer #2 (Remarks to the Author):

Thank you for addressing the concerns raised. I think this report will be a meaningful contribution to the field of smoking and its associated predictors in diverse settings globally.

- We are glad to hear that our revision addressed all the concerns raised. We also appreciate the reviewer recognizing our study’s potential contribution to the smoking literatures.

Point-by-point response to reviewers' comments

Reviewer #1 (Remarks to the Author):

My concern regarding the statistical approach for Table 2 remains. As I mentioned in the previous review, Table 2 shows the results the continuous daily cigarette measure with the total sample (N = 202,898). Given the small smoker sample (N = 38,290), I still believe it is inappropriate to run a model combining a large number of zeros (81% - no daily cigarettes) with positive values (19%).

When comparing the results in Table 2 with those in Table 5, I notice differences. I do not completely agree with your explanation regarding potential sources of differences in some predictors. I believe the model used for Table 2 is mis-specified, leading to biased estimates. I suggest that the models should include a binary outcome using the total sample, and then a continuous outcome using the smoker sample.

- Thank you for carefully reviewing our work and providing guidance on how our results presented in Table 2 are potentially biased.
- We have updated the manuscript by removing Table 2 from the main text, while we included the table in the online supplement (Table S47) for transparency in reporting because the total sample analysis was part of our preregistration (<https://osf.io/6umhp>). We have also updated Table 3 by removing sensitivity analysis results associated with Table 2 from the table and present them in the online supplement (Table S48) for the same reason.
- We have revised the method section to note how the results in Tables 4 and 5 (which are now Tables 2 and 4, respectively) were our attempt at approximating a zero-inflated model given the constraints of our analytic strategy in the GFS (see pp. 7-8).
- We have also made changes throughout the manuscript accordingly, mostly in the results section (pp. 14-26; see also the abstract and ps. 3, 11, 27, 28, 29, and 32).
- Again, thank you for your comments and help ensuring our work is the best possible.

Point-by-point response to Reviewer 1' comments

Reviewer #1 (Remarks to the Author):

I appreciate the scope and scale of this study, particularly the extensive data collected across 22 countries. I recognize the value of such a large, multi-country dataset and the effort involved in the analyses. However, I am having difficulty fully understanding or endorsing the authors' interpretations of the results.

With the revised approach and analysis results, the authors presented the estimated effects of the same set of predictors on two outcomes: (1) the likelihood of being a daily smoker and (2) the number of cigarettes smoked daily. However, I am having difficulty understanding how the authors interpreted the estimated effects, particularly for certain predictors. In several cases, their interpretations appear to be overstated or not well-supported by the reported estimates.

- We appreciate the reviewer raising this issue, which allowed us to revise our interpretations of certain predictors' estimated effects by correcting their overstatement and inconsistency.

Specifically, I am trying to understand the interpretation of relative risks (RRs) presented in Table 2. Take, for example, the predictor "religious service attendance at age 12." The reported RRs for the three comparison categories relative to "never" attending religious services were approximately 1.0 (ranging from 0.96 to 1.05 – weak effects), suggesting little to no association with the outcome. The proportion of effects below and above the defined threshold were 32% and 14%, respectively – interpreted as roughly 7 countries showing RRs below the threshold and 3 above, out of 22 countries. The remaining 12 countries appear to have RRs close to 1.00, indicating no strong association with religious service attendance at age 12. Is this interpretation correct?

- Generally agreeing with the reviewer's interpretation, we revised the paragraph where we discussed the estimated effects of religious service attendance at age 12 on daily smoking in adulthood (p. 19). Specifically, we explicitly stated that the effects were mixed and weak, saying that the observed weak effect was confined to the category of most frequent attendance.
- We also added the cross-country variation in the effect of "at least once a week" religious service attendance, which the reviewer mentioned, while keeping its discussion in the original place (p. 28).

If so, I am unsure how the authors concluded that religious service attendance was a protective factor against daily smoking. This interpretation seems inconsistent with the reported RRs and the proportions of effects crossing a meaningful threshold.

- We removed the description of childhood religious service attendance being a protective factor for adult daily smoking in presenting the total sample results.

Age 12 religious service attendance (Copied from Table 2)

(Ref: Never) Estimated proportion of effects by threshold

At least 1/week 0.96 (0.90,1.03) 0.32 < 0.90 > 1.10

1-3/month 1.05 (0.95,1.16) 0.14 0.36 75.7

< 1/month 1.00 (0.95,1.05) 0.05 0.05 30.5

I encountered similar issues with the authors' interpretation of other predictors. For example, in the models of binary smoking, the reported effects for socioeconomic status (SES) and health were not consistently strong or unidirectional. In the models for daily cigarette consumption, the effects of parental marital status, some SES categories, health status, outsider status, and immigration background were also mixed or weak in magnitude. Overall, the direction and strength of the estimated effects for these predictors do not clearly support the conclusions drawn in the Abstract's Results section. As such, I do not fully agree with the authors' interpretations.

- Agreeing with the reviewer's first point, we revised our interpretations of the effects of childhood family SES and physical health on the binary measure of daily smoking (pp. 17-18).
- To address the second point, we revised interpretations of "the effects of parental marital status, some SES categories, health status, outsider status, and immigration background" on the quantity of daily cigarette consumption among smokers to clearly indicate how they were mixed in direction and strength as well as "weak in magnitude" (ps. 22 & 24; see also p. 25).
- In response to the third point, we revised not only the Abstract's Results section but also a paragraph summarizing overall results in the Discussion section (pp. 27-28).